# Long-term persistence of viral RNA and inflammation in the CNS of macaques exposed to aerosolized Venezuelan equine encephalitis virus

Henry Ma[1,2◉], Joseph R. Albe[1◉], Theron Gilliland[1], Cynthia M. McMillen[1,2], Christina L. Gardner[1], Devin A. Boyles[1], Emily L. Cottle[1], Matthew D. Dunn[1], Jeneveve D. Lundy[1], Noah Salama[1,2], Katherine J. O'Malley[1], Ivona Pandrea[2,3], Tobias Teichert[4,5], Stacey Barrick[1], William B. Klimstra[1,6‡], Amy L. Hartman[1,2‡], Douglas S. Reed[1,6‡]*

1 Center for Vaccine Research, School of Medicine, University of Pittsburgh, Pittsburgh, Pennsylvania, United States of America, 2 Department of Infectious Diseases and Microbiology, School of Public Health, University of Pittsburgh, Pittsburgh, Pennsylvania, United States of America, 3 Department of Pathology, School of Medicine, University of Pittsburgh, Pittsburgh, Pennsylvania, United States of America, 4 Department of Psychiatry, University of Pittsburgh School of Medicine, Pittsburgh, Pennsylvania, United States of America, 5 Department of Bioengineering, University of Pittsburgh, Pittsburgh, Pennsylvania, United States of America, 6 Department of Immunology, School of Medicine, University of Pittsburgh, Pittsburgh, Pennsylvania, United States of America

◉ These authors contributed equally to this work.
‡ WBK, ALH, and DSR also contributed equally to this work as senior authors.
* dsreed@pitt.edu

**Data Availability Statement:** All relevant data are within the manuscript and its Supporting Information files.

## Abstract

Venezuelan equine encephalitis virus (VEEV) is a positively-stranded RNA arbovirus of the genus *Alphavirus* that causes encephalitis in humans. Cynomolgus macaques are a relevant model of the human disease caused by VEEV and are useful in exploring pathogenic mechanisms and the host response to VEEV infection. Macaques were exposed to small-particle aerosols containing virus derived from an infectious clone of VEEV strain INH-9813, a subtype IC strain isolated from a human infection. VEEV-exposed macaques developed a biphasic fever after infection similar to that seen in humans. Maximum temperature deviation correlated with the inhaled dose, but fever duration did not. Neurological signs, suggestive of virus penetration into the central nervous system (CNS), were predominantly seen in the second febrile period. Electroencephalography data indicated a statistically significant decrease in all power bands and circadian index during the second febrile period that returned to normal after fever resolved. Intracranial pressure increased late in the second febrile period. On day 6 post-infection macaques had high levels of MCP-1 and IP-10 chemokines in the CNS, as well as a marked increase of T lymphocytes and activated microglia. More than four weeks after infection, VEEV genomic RNA was found in the brain, cerebrospinal fluid and cervical lymph nodes. Pro-inflammatory cytokines & chemokines, infiltrating leukocytes and pathological changes were seen in the CNS tissues of macaques euthanized at these times. These data are consistent with persistence of virus replication and/or genomic RNA and potentially, inflammatory sequelae in the central nervous system after resolution of acute VEEV disease.

**Funding:** All authors (HM, JRA, TG, CMM, CLG, DAB, ELC, MDD, JDL, NS, KJO, IP, TT, SB, WBK, ALH, DSR) listed received funding for these studies under grant W911QY-15-1-0019, funded by the Defense Threat Reduction Agency (https://www.dtra.mil/). The sponsors reviewed the manuscript before publication but did not play a role in study design, data collection, analysis, or the decision to publish.

**Competing interests:** The authors have declared that no competing interests exist.

## Author summary

Although naturally transmitted by mosquito, Venezuelan equine encephalitis viruses (VEEV) can be highly infectious when aerosolized. In humans, VEEV are only rarely fatal but cause a severe, biphasic fever with neurological symptoms including severe headache, a stiff neck, and photophobia. We report here our efforts to further characterize the disease caused by VEEV in the cynomolgus macaque, using an infectious clone of a human VEEV isolate, to explore the long-term effects of VEEV infection, and the utility of radio-telemetry in continuous monitoring of electroencephalography and intracranial pressure to explore the relationship between fever, virus penetration of the brain, and neurological disease.

## Introduction

Venezuelan equine encephalitis virus (VEEV) is one of the three encephalitic alphaviruses found in the Americas that are naturally mosquito-transmitted but laboratory accidents have shown it can also cause disease when inhaled [1–5]. Enzootic serotypes of VEEV are found throughout Central and South America. Throughout the 20th century, epizootic serotypes emerged that have caused major outbreaks in both equine and human populations. While lethal in unvaccinated equines, in humans, the disease caused by VEEV is typically only fatal in the very young or very old [1]. Typical human disease is characterized by a biphasic fever, nausea, stiff neck, photophobia, and a severe headache. Laboratory findings can include lymphopenia and viremia.

Because alphaviruses can be highly infectious when aerosolized, there is concern that they could be used as bioweapons. Both rhesus and cynomolgus macaques have been used as non-human primate models of the human alphavirus disease [6]. Studies in rhesus macaques demonstrated that trafficking of the virus via the olfactory nerves was important for CNS disease after aerosol infection [7,8]; however, the majority of vaccine studies have used cynomolgus macaques. Wild-type VEEV is generally lethal in mice, but in both rhesus and cynomolgus macaques, VEEV disease is typically self-limiting and the animals eventually recover, much like the majority of infected humans. Fever, viremia, and lymphopenia are the major clinical signs in macaques, although limited signs suggestive of neurological disease (loss of coordination, dehydration, anorexia, slight tremors) have also been noted [9].

Most previous work in macaques used virus isolates of uncertain passage history or which were not originally isolated from human cases. The Trinidad Donkey (TrD) IA/B epizootic isolate of VEEV, which was isolated from a donkey in the mid-twentieth century, has been the prototypical epizootic strain used to evaluate potential vaccines and therapeutics since the 1960s [10]. However, the Food & Drug Administration's (FDA) Animal Rule for licensure of medical countermeasures requires the use of a well-characterized low-passage isolate from a human patient in pivotal efficacy studies [11]. INH-9813, a human isolate of the IC epizootic subtype of VEEV, is a VEEV isolate that meets this requirement. An infectious cDNA clone has been created for use in animal studies to preserve virus sequences such that further cell culture adaptive mutations are not accumulated and studies can be easily standardized between laboratories [12]. Prior to use in efficacy studies, however, it was critical to demonstrate that the disease caused by the clone of INH-9813 is similar to what has been reported previously in nonhuman primates (NHP) with other isolates, and further, to establish the relevance of this model to human disease. If the cynomolgus macaque is to be used for evaluating potential

therapies (as opposed to vaccines), biomarkers indicative of different disease stages are needed so that stage-appropriate therapeutic treatment can be initiated. Also, long term consequences of infection such as persistence of viral RNA and/or host inflammatory responses have not been investigated in previous studies but represent a potential indicator of disease severity.

Previous studies evaluating VEEV in NHP models, particularly via inhalation, have primarily focused on fever, viremia, and lymphopenia as the predominant clinical signs used to assess disease [7–10,13–15]. Neurological disease was only assessed by twice-daily clinical observation of animal behavior; signs suggestive of central nervous system disease in macaques infected with VEEV are typically mild and could be mistaken for other signs (e.g., mild tremors could be shivering from fever). While this has been sufficient for evaluation of vaccine efficacy, better characterization of the disease in macaques, particularly penetration of the virus into central nervous system, was essential for adapting the macaque as a model for use in evaluating compounds that might be useful for post-exposure prophylaxis or therapeutic use. Particularly desirable was a 'trigger' to initiate treatment similar to what has been done to license antibiotics for the treatment of pneumonic plague [16]. One question was whether other indicators besides fever could be used as that trigger. The purpose of this study therefore was three-fold: to evaluate the disease caused by inhalation of INH-9813, a human isolate of VEEV selected for pivotal animal studies, generated from an infectious cDNA clone; to better characterize the cynomolgus macaque model of VEEV and characterize the long-term effects of infection; and to evaluate the use of radiotelemetric measurement of electroencephalography (EEG) and intracranial pressure (ICP) for detecting CNS disease in addition to the febrile response.

## Materials and methods

### Ethics statement

The animal work performed adhered to the highest level of humane animal care standards. The University of Pittsburgh is fully accredited by the Association for Assessment and Accreditation of Laboratory Animal Care (AAALAC; accreditation unit number 000496). All animal work was performed under the standards of the Guide for the Care and Use of Laboratory Animals published by the National Institutes of Health and according to the Animal Welfare Act guidelines (OLAW assurance number D16-00118; USDA registration number 23-R-0016). All animal studies adhered to the principles stated in the Public Health Services Policy on Humane Care and Use of Laboratory Animals. The University of Pittsburgh Institutional Animal Care and Use Committee (IACUC) and the Department of the Army's Animal Care and Use Review Office (U.S. Army Medical Research and Materiel Command [USAMRMC]) approved and oversaw the animal protocols for these studies, protocol 17100664.

### Biological safety

All work with VEEV was conducted in the University of Pittsburgh Center for Vaccine Research (CVR) and the Regional Biocontainment Lab (RBL), which is currently registered and approved for work with VEEV by the Federal Select Agent Program. All personnel have undergone appropriate Security Risk Assessment (SRA) and are approved through the University of Pittsburgh suitability assessment program. Respiratory protection for all personnel when handling infectious samples or working with animals is provided by powered air-purifying respirators (PAPRs; Versaflo TR-300; 3M, St. Paul, MN). Aerosol exposures were conducted within a class III biological safety cabinet (BSC). Liquid and surface disinfection was performed using Vesphene IIse detergent (dilution, 1:128; Steris Corporation, Erie, PA), while solid wastes, caging, and animal wastes were steam sterilized in an autoclave.

## Challenge virus

Macaques were challenged via aerosol with VEEV alphavirus strains INH9813 or TrD [17], derived from cDNA clones [12,18–20]. Viruses were produced initially by full-length, capped genome synthesis from linearized cDNA clones (mMessage Machine, Thermofisher) and electroporation of 10–20 μg of RNA into BHK 21 cells. Supernatants were harvested at 18–24 hours after electroporation and used to infect 1900 cm2 roller bottle cultures of Vero cells. At 18–24 hours after infection, Vero cell supernatants were harvested and subjected to low speed clarification prior to loading onto a 20%/60% discontinuous sucrose gradient in TNE buffer (10 mM Tris, 10 mM EDTA, 2 M NaCl, pH 7.4) and centrifugation for 3.5 hours at 24,000 RPM. The virus-containing interface was then harvested and virus particles were pelleted over a 20% sucrose (in TNE) gradient for 18–24 hours at 24,000 rpm. Pelleted virus was resuspended in Opti-MEM (Gibco 31985–070) and stored at -80˚C. Individual lots of virus were tested for virulence by aerosol dose-step infection of CD-1 mice and approved for NHP use only after virulence in mice was verified ($LD_{50}$ <1 pfu, time to death 5–7 days). INH-913 was sequenced by Sanger sequencing of overlapping RT-PCR amplicons. No differences with the cDNA clone sequence were observed. Coverage did not include the 5' and 3' 20 nucleotides of the stock virus reflecting primer binding sites (see S2 Table and S1 Data for primers used and sequence data).

## Cells & media

For plaque assays and neutralizing titer assays, BHK (ATCC CCL-10) and Vero (ATCC CCL-81) cells were obtained from the American Type Culture Collection. BHK cells were maintained in Roswell Park Memorial Institute 1640 media (RPMI) supplemented with 10% donor bovine serum (DBS), 10% tryptose phosphate broth, 200mM L-glutamine (L-Glut; Sigma), 10,000 units/mL penicillin (Sigma) and 10mg/ml streptomycin (Sigma). Vero cells were maintained in Dulbecco's modified Eagle's medium (DMEM) supplemented with 10% fetal bovine serum (FBS), 200mM L-glutamine (L-Glut; Sigma), 10,000 units/mL penicillin (Sigma) and 10mg/ml streptomycin (Sigma). For plaque assays, BHK cells were treated with Trypsin-EDTA for 5 minutes, washed, and seeded into 6-well plates at least 24 hours before use.

## Plaque assay

The presence of virus in blood and nasal swabs was determined by standard alphavirus plaque assay on BHK cells at each sampling time [21]. Briefly, sera were diluted serially in 10-fold increments into phosphate buffered saline (PBS) with calcium/magnesium additive and 1% donor calf serum and then used to infect monolayers of BHK cells for 1 hour at 37˚C. Monolayers were then overlaid with 0.5% immunodiffusion agarose (MP Bio 952012) in BHK cell growth medium and plates were incubated at 37˚C for 24–48 hours. Monolayers were stained after ~24–36 hours with 0.3 mM neutral red (Acros Organics 229810250) in PBS with 1% donor calf serum and plaques were enumerated.

## General animal procedures

Cynomolgus macaques ranged from 2–9 years of age and included both male and female animals. Prior to use, the macaques were verified to be serologically negative for alphaviruses as well as Herpes B Virus, simian immunodeficiency virus, simian T-cell leukemia virus, and simian retrovirus. Macaques were implanted with telemetry devices as described below. As needed, macaques were sedated for phlebotomy with 10mg/kg ketamine administered via intramuscular injection using a safety needle and blood collected from the femoral or

saphenous vein. For euthanasia, macaques were sedated with 20mg/kg ketamine followed by injection of sodium nitroprusside mixed with 12 ml of NaCl, followed by 200 mg/kg of sodium pentobarbital (Patterson Veterinary). Once euthanasia was confirmed via cessation of heart beat, macaques were perfused via the left ventricle with saline. Tissues were inactivated in 10% neutral buffered formalin. Paraffin-embedded tissues were cut, mounted on slides, and stained with H&E for pathology interpretation.

## Telemetry implant surgery

Each macaque was equipped with a PhysioTel Digital radiotelemetry implant (Data Sciences International, DSI, St. Paul, MN) capable of continuously transmitting EEG, ICP, and temperature. Prior to implantation surgery, each macaque was anesthetized by injection of ketamine hydrochloride (20 mg/kg) and atropine (0.4 mg/kg); once anesthesia was confirmed they were maintained on ~1.5% isoflurane gas anesthesia for the duration of surgery. Each macaque received an intravenous (IV) catheter in the greater saphenous vein for 3% normal saline, a endotracheal tube for intubation, with continuous pulse oximetry and rectal thermometry for vital sign monitoring. Prior to draping and fixture into stereotaxic apparatus, the head, neck and upper back of each macaque was shaved and scrubbed in triplicate with betadine and chlorhexidine. Surgical implantation of the implant was performed as previously described [18]. Post-surgery, macaques were given analgesia and observed until recovered. At least 2 weeks after surgery macaques were transferred to the ABSL-3 facility.

## Aerosol exposure

Aerosol exposures were performed under the control of the Aero3G aerosol management platform (Biaera Technologies, Hagerstown, MD) as previously described [22]. Macaques were anesthetized with 6 mg/kg Telazol (Tiletamine HCl / Zolazepam HCl); once anesthesia was confirmed the macaque was weighed, bled, and transported to the Aerobiology suite using a mobile transport cart. The macaque was then transferred from the cart into a class III biological safety cabinet and the macaque's head was placed inside a head-only exposure chamber. Jacketed External Telemetry Respiratory Inductive Plethysmography (JET-RIP; DSI) belts were placed around the upper abdomen and chest of the macaque and calibrated to a pneumotach. This allowed monitoring and recording of respiratory function during the exposure via the Ponemah software platform (DSI) during the aerosol [23]. Exposures were 10 minutes in duration. Aerosols were generated using an Aerogen Solo vibrating mesh nebulizer (Aerogen, Chicago, IL) as previously described, with a total airflow of 16 lpm of air into the chamber (one complete air change every 2 minutes) [24]. Total exhaust from the chamber including sampling was equal to intake air. To determine inhaled dose, an all-glass impinger (AGI; Ace Glass, Vineland, NJ) was attached to the head-only chamber and operated at 6 lpm, -6 to -15 psi. The AGI is attached in the breathing zone of the animal. Particle size was measured once for 30 seconds during each exposure at 5 minutes using an Aerodynamic Particle Sizer (TSI, Shoreview, MN). A 5-minute air wash followed each aerosol before the macaque was removed from the cabinet, and transported back to its cage and observed until fully recovered from anesthesia. Virus concentration in nebulizer and AGI samples was assessed by plaque assay; inhaled dose was calculated as the product of aerosol concentration of the virus and the accumulated volume of inhaled air [25]. Mock-infected macaques were exposed for 10 minutes to cell culture media.

## EEG/ICP/Temperature data acquisition

Signals from DSI implants were transmitted to TRX-1 receivers connective via a communication link controller (CLC) to a computer running Ponemah v5.2 or 5.3 (DSI). Closed circuit

cameras (Axis Model No. M-1145) were positioned to continuously record macaque behavior including neurological abnormalities such as seizures and sleep disruption. Camera video was recorded through a Ponemah interface with MediaRecorder software (Noldus Information Technology, Leesburg, VA). Telemetry and video data was collected continuously from a baseline period (at least two days preceding aerosol challenge) until necropsy. At least once daily, typically just after morning observations, data acquisition was stopped for a maximum of 15 minutes, data was transferred to a network server, and then acquisition was restarted.

## Temperature data analysis

Temperature data collected via radiotelemetry was exported from Ponemah as 15-minute averages into an Excel spreadsheet. Data was checked for missing or erroneous data points (body temperature $<27°C$ or $>43°C$) and analyzed using an auto-regressive integrated moving average (ARIMA) model in MATLAB R2019A (MathWorks, Natick, MA). Prior to challenge, for each macaque the baseline temperature data was used to generate hourly temperature ranges for use in clinical scoring to determine significant deviations in temperature. To quantify fever responses, predicted temperatures are subtracted from actual temperatures using the ARIMA model (residual temperature); a significant elevation in temperature (fever) was defined as a residual temperature greater than the upper limit from predicted, which was calculated as 3 multiplied by the square root of the residual sum of squares. Maximum temperature deviation is defined as the highest residual temperature elevation after challenge. Fever duration is calculated as the number of points after challenge above the residual upper limit divided by 4 to convert to hours. Fever-hours is the sum of the points after challenge above the residual upper limit divided by 4 to convert to hours. Because the EEG/ICP implants were subcutaneous on the upper back of the macaque, temperature readings were lower than would be expected for core body temperatures but still exhibited 1–2°C diurnal variation prior to challenge. A new version of our temperature analysis code was written for VEEV-infected macaques to quantify fever responses during the two febrile periods observed between days 0.5–2 and 2.5–8 dpi as well as the incubation (0–0.5 dpi) and recovery (8+ dpi) periods. The code used is available on GitHub at https://github.com/ReedLabatPitt/Reed-Lab-Code-Library.

## EEG Data analysis

Analysis of EEG and ICP data was performed as previously described [26]. Briefly, Ponemah data was opened in NeuroScore (v.3.0.0, DSI) software package, which facilitated batch processing of data to European Data Format (.edf) for analysis. EEG data was analyzed using in-house MATLAB (Mathworks, 2018a) scripts building on routines from the EEGLAB MATLAB toolbox EEGLAB [27]. Four distinct frequency bands were defined: delta [0.5-4Hz]; theta [4-8Hz]; alpha [8-12Hz]; and beta [12-30Hz] and estimated power in each band as the average power of all frequencies in the corresponding range. The EEG power spectral density magnitudes for each macaque were converted to percent change from baseline values across the whole of the specified periods: baseline, pre-symptomatic (0–2 DPI, inclusive), febrile (3–6 DPI, inclusive), and recovery (6+ DPI). Group averages of percent change at discrete frequencies were taken for severe and nonsevere cohorts of macaques and compared against baseline by repeated measures ANOVA.

## Circadian index construction

To visualize changes in the circadian patterning of the time-resolved EEG power bands, a circadian index was constructed by subtracting the normalized beta power band from the

normalized delta power band. Analysis of the periodicity of the circadian index was hampered by segments of missing data; to overcome this limitation, the Lomb-Scargle methodology of least-squares frequency analysis was employed to account for the gaps [28]. Circadian indices were compared via repeated measures ANOVA of their fundamental frequencies, or the lowest frequency of the associated waveform.

### ICP Data analysis

ICP data was exported from Ponemah into MATLAB as time series data and grouped into mean daily averages to minimize noise; measurements were converted into percentage change from baseline, and repeated measures ANOVA was used to assess results for statistical significance.

### Clinical scoring

Macaques were scored twice daily by direct observation, at least six hours apart. The scoring system used requires observations be increased for macaques that developed severe disease but this was not required for VEEV, which only rarely causes lethal disease in macaques. The approved clinical scoring system consisted of three scoring categories. *Neurological score*: 1 = normal, 2 = occasional loss of balance or muscle control (ex: stumbling, unsteadiness), 3 = occasional nystagmus (eye oscillation/twitching), head pressing, tremors, 4 = loss of balance, nystagmus, head pressing, tremors; occasional seizures, 5 = frequent seizures, 6 = comatose (prompt euthanasia). *Activity score*: 1 = normal: responds to observer entering the room, frequent eye contact and body language interactions (both positive or aggressive postures/ expressions); 2 = less active: responds to observer approaching the cage; body language interactions less frequent or intense; 3 = sluggish: only responds when prodded or when observer rattles the cage, maintains hunched posture with back to observer; limited eye contact and body language interactions; glassy eyes, grimace or sad facial expression; 4 = upright but inactive; does not respond to observer rattling the cage, ignores all stimuli; 5 = recumbent/moribund (prompt euthanasia). *Temperature score*: 1 = Normal = baseline to 1.5 degrees above baseline; 2 = Mild Fever = 1.6–3.0 degrees above baseline; 3 = Moderate Fever = 3.1 to 4.0 degrees above baseline; 4 = Severe Fever = >4.0 degrees above baseline or Mild Hypothermia = 0.1–2 degrees below baseline; 5 = Moderate Hypothermia = 2.1–5 degrees below baseline; 6 = Severe Hypothermia = >5 degrees below baseline (prompt euthanasia). The daily clinical score for each macaque was the sum of the scores for these 3 criteria. Based on this scoring system, a healthy macaque's baseline score was 3. A total score of 10 (3+3+4 on the 3 categories above) warranted observations every eight hours, including overnight. Macaques that reach a cumulative score of 14 were promptly euthanized. Email alerts on the Ponemah software were also used to notify study personnel of fever and severe hypothermia.

### Whole blood processing

Animals were sedated for sampling after infection by administration of 10 mg/kg ketamine via intramuscular injection. Once sedated, 2–3 ml of blood was drawn from either the right or left femoral vein into an EDTA tube. To obtain plasma, the tube was centrifuged at 600x g for 5 minutes. Plasma was then aliquoted and frozen for serologic, immunologic and virologic assays. CBC analysis was performed using the Abaxis HM5 hematology analyzer. Blood chemistry analysis was performed using the Comprehensive Diagnostic Panel rotor (Abaxis 500– 0038) on an Abaxis VS2 chemistry analyzer.

## Brain cell isolation

After perfusion of macaque with saline, brains were removed, weighed and used for cell isolation as described [29]. For each brain, 1 g pieces dissected from the frontal, temporal, occipital, parietal, cerebellar regions were covered with digestion buffer consisting of modified HBSS without calcium and magnesium, 10mg/ml DNase I (Sigma 10104159001), 20mg/ml of collagenase (Sigma C2674) and mechanically digested using a scalpel. The sample was then incubated at 37˚C for 45 minutes on a continuous rocker. Every 15 minutes, the sample was mechanically triturated using a serological pipet. The resulting homogenate was then filtered through a 40 μm cell strainer and washed twice with wash buffer, consisting of HBSS with 3% FBS and 10 mg/ml DNase I and centrifuged at 500 x gravity for 8 minutes at room temperature. The supernatant was removed and the remaining pellet was suspended in 80% stock isotonic Percoll (SIP) (Sigma GE17-0891-01) made in HBSS solution. The suspension was subsequently overlaid with 10 ml of 38% SIP, 10 ml 21% SIP, followed by 5 ml HBSS with 3% FBS and centrifuged at 480 x gravity for 35 minutes, no brake. The third interface was removed, washed twice with modified HBSS containing 3% FBS and then suspended in 1 ml FACS buffer. Cells were counted using a hemocytometer and placed on ice. Cells suspended in FACS buffer were placed in a V-bottom 96-well plate and centrifuged for 500 x gravity for 4 minutes at 4˚C. An Fc block was performed by adding 2μL of purified anti-CD32 (BD Biosciences) and 18μL of FACS buffer per sample for 20 minutes on ice in the dark. Cells were then washed twice with 200uL FACS buffer and stained with live/dead. Cells were washed with 200μL of FACS buffer and then stained antibody mix for 30 minutes on ice in the dark. Samples that required intracellular stain were permeabilized and fixed using BD Cytofix/Cytoperm (BD554714) and then washed with FACS perm-buffer. The stained samples were then washed twice with 200μL FACS buffer followed by fixation with 200μL of 4% PFA. Samples were run on a BD LSRII and analyzed using FlowJo 10.5.0.

## Cytokine analysis

Cytokine analysis was performed on plasma, CSF, and brain samples using Biolegend LEGENDplex 13-plex kit (Cat. No. 740389). Lethal brain samples were diluted 1:2 in assay buffer, plasma was run at 1:4 dilution, and convalescent brain or mock-infected animals as well as all cerebral spinal fluid were run undiluted. Samples were prepared following LEGENDplex protocol for v-bottom kit. Samples were then run on FACSAria flow cytometer immediately following staining. Results were then analyzed using LEGENDplex Data Analysis Software. Analytes measured were interleukin (IL)-6, IL-10, IP-10, IL-β, IL-12p40, IL-17A, IFN-β, IL-23, TNF-α, IFN-γ, GM-CSF, IL-8 and MCP-1. An ELISA assay was used to measure MMP9 levels in plasma (R&D Systems; DMP900).

## Tissue extraction and processing

Whole brain sections (frontal, parietal, temporal, occipital, cerebellum, thalamus), cervical lymph nodes (CLN), serum, cerebrospinal fluid (CSF), and nasal swabs were collected for VEEV control macaques. For whole brain sections and CLN, 100mg of tissue was harvested, suspended in 900 μL Tri-Reagent (Invitrogen), then homogenized using Omni tissue homogenizer (Omni International). For liquid samples, 100 μL of the specimen was suspended in 900μL Tri-Reagent, then thoroughly mixed by inversion. After a 10-minute incubation to ensure virus inactivation, the samples were transferred to a fresh tube prior to removal from the BSL-3 facility. Subsequent storage at -80˚C or RNA isolation, cDNA synthesis, and qPCR analyses occurred in a Select Agent BSL-2 setting.

## RNA Isolation

RNA isolation was performed on whole tissue homogenate samples using a modified Invitrogen PureLink Viral RNA/DNA kit protocol. In brief, 200 μL of chloroform was added to the inactivated tissue homogenate/Tri-Reagent mixture, inverted vigorously for 30 seconds then centrifuged at 4˚C at 12,000 RCF for 15 minutes to separate the organic phase from the RNA-containing aqueous phase. The aqueous phase was collected, mixed with an equal volume of 70% ethanol, and then applied to the PureLink spin column. For the remainder of the RNA isolation procedure, the Purelink Viral RNA/DNA kit protocol was used, including deoxyribonuclease treatment. RNA was eluted in RNase-free water to a final volume of 40 μL then stored at -80˚C until further processing into cDNA.

For serum, CSF, and nasal swab samples, 5 μL of polyacryl carrier (Molecular Research Center) was added to the specimen/Tri-Reagent mixture, and incubated at room temperature for 30 seconds. Subsequently, 200 μL of chloroform was added to the mixture and incubated at room temperature for an additional 3 minutes, then centrifuged at 4˚C at 12,000 RCF for 15 minutes. The aqueous phase was collected and then combined with an equal volume of 100% isopropanol. The mixture was incubated at room temperature for 10 minutes before centrifugation at 4˚C at 12,000 RCF for 10 minutes. The supernatant was removed and 1mL of 70% ethanol was added before further centrifugation at room temperature at 12,000 RCF to pellet the RNA. The supernatant was removed, and the pelleted RNA was resuspended in nuclease-free water. RNA was eluted in RNase-free water to a final volume of 40μL then stored at -80˚C until further processing into cDNA.

## Complementary DNA (cDNA) Synthesis

cDNA synthesis was performed using the First-strand cDNA Synthesis M-MLV Reverse Transcriptase Kit (Invitrogen) protocol, including RNaseOUT (Invitrogen). 5 μL of whole tissue samples (20 ng/μL) or undiluted plasma, CSF, or nasal swab RNA was added to the cDNA synthesis reaction. For quantification of VEEV vRNA, the following cDNA synthesis primer with a T7 promoter tag (bolded) targeting the positive strand was used: VEEV (INH9813) 5'-**GCGTAATACGACTCACTATA**GTCTTCTGTTCACAGGTACTAGAT -3' (Integrated DNA Technologies, IDT). Primers targeted the VEEV NSP2 coding region. For quantification of cytokines, chemokines, or proteases, random oligonucleotide primers (ThermoFisher) were used to reverse transcribe all RNA using the First-strand cDNA Synthesis M-MLV Reverse Transcriptase Kit protocol. Thermocycler parameters recommended by the manufacturer were utilized. cDNA was stored at -80˚C until further analysis via qPCR.

## qRT-PCR

For quantitation of genomic vRNA, qRT-PCR was performed using the 2x Fast TaqMan Universal PCR Master Mix, No AmpErase UNG (Applied Biosystems), following the manufacturer's instructions. Forward primer and probe targeting VEEV include: forward primer 5'-CCGGAAGAGTCTATGACATGAA-3' and probe 5'- CTGGCACGCTGCGCAATTA TGATC -3' (IDT). The probe was labeled at the 5' end with the reporter molecule 6-carboxy-fluorescein (6-FAM) and quenched internally at a modified thymidine residue with Black Hole Quencher (BHQ1), with a modified 3' end to prevent probe extension by Taq polymerase. Reverse primer for VEEV targeted the T7 promoter tag generated during cDNA synthesis: 5'-GCGTAATACGACTCACTATA-3' (IDT). Thermocycler (QuantStudio 6 Real-Time PCR Flex System; Applied Biosystems) parameters consist of initial denaturing: 95˚C for 20 seconds, and cycling PCR amplification (45 cycles): 95˚C for 3 seconds and 60˚C for 20 seconds. Quantitation of virus was determined by comparing the cycle threshold (CT) values from

unknown samples to CT values from positive-sense VEEV vRNA standard curve from 1:10 serial dilutions. Positive-sense vRNA was developed in-house by in vitro transcription, using the mMessage mMachine T7 kit (Ambion) and following the manufacturer's instructions [30]. Limit of detection (LOD) determination was carried out in accordance with the Minimum Information for the Publication of Quantitative Real-Time PCR Experiments guidelines [31]. LOD for VEEV qPCR was 488 genome copies. The LOD for VEEV qPCR, $3.9 \times 10^4$ genome copies/mL or mg, was determined based on the lowest amplified qPCR standard and the background amplification detected in mock-infected controls and baseline (0dpi) samples. Background amplification detected in mock-infected controls and baseline (0dpi) samples were subtracted from corresponding infected samples and later time-points. Viral RNA titers shown are based on total viral copies above LOD.

## Semi-quantitative RT-PCR

2x Fast Taqman Universal PCR Master Mix, No AmpErase UNG and Taqman Gene Expression Assay kits (Invitrogen) were used for semi-quantitative analysis of cytokines, chemokines, proteases, and traumatic brain injury markers (MMP-9, GFAP, LIF, IL-1β, MCP-1, IP-10, IFNγ. IL-6, IL-8) within whole brain tissues, following manufacturer's instructions. Probes were labeled at the 5' end with 6-FAM and quenched with BHQ1, with a modified 3' end to prevent probe extension by Taq polymerase. Thermocycling (QuantStudio 6 Real-Time PCR Flex System; Applied Biosystems) parameters comprised the following: hold step, 50˚C for 120 seconds; initial denaturing, 95˚C for 120 seconds; and cycling PCR amplification, 95˚C for 1 second and 60˚C for 20 seconds (40 cycles). All kits were targeted to cynomolgus macaque-specific, highly-conserved amplicons that spanned one or more exons to guarantee desired mRNA transcript specificity. Endogenous controls comprised TaqMan Gene Expression Assay kits (Invitrogen) for GAPDH and β-actin, with corresponding brain tissue from a mock-infected macaque serving as the reference tissue. Inter-run calibration to normalize inter-run/inter-plate variability was carried out according to best practices [31,32].

## Pathology

Tissue samples were fixed with 10% neutral buffered formalin according to approved inactivation protocols. Upon removal from BSL-3, samples were embedded in paraffin blocks and sectioned onto slides by the Histology Lab in the McGowan Institute of Regenerative Medicine. Slides were stained with hematoxylin and eosin, followed by pathology scoring. Slides were assessed for leukocyte infiltration (including meningitis and perivascular leukocyte infiltration), hemorrhage, and appearance of neurons. Lesions were scored as 0 = normal, 1 = mild, 2 = moderate, 3 = severe, 4 = widespread and severe.

## Statistics

GraphPad Prism was used for statistical determinations. GraphPad Prism and MATLAB were used to analyze the EEG and ICP data sets for statistical significance.

## Results

### Fever response

A total of 14 cynomolgus macaques (8 male; 6 female) were exposed to VEEV INH 9813 by head-only inhalational infection using a small particle aerosol delivery system and vibrating mesh nebulizer described previously [24]. Animals were infected with a range of doses

spanning 6.0–8.4 1og$_{10}$ pfu (S1 Table). Two additional macaques were exposed to 8.3 and 7.6 log$_{10}$ pfu of VEEV TrD strain [17,33]. Six-hour median residual temperatures for individual macaques revealed a generally biphasic response at all exposure doses (Fig 1). For display purposes, macaques exposed to INH-9813 were split into "low" (6–6.9 log$_{10}$ pfu), "mid" (7.0–7.3 log$_{10}$ pfu), and "high" (7.5–8.4 log$_{10}$ pfu) dose groups (n = 6/group). S1 Table shows maximum residual temperatures, fever duration and fever severity overall and for each febrile period in individual macaques. With both INH9813 and TrD (Fig 1; square symbols) strains at all the doses tested, there was a significant spike (typically ≥2˚C from predicted) in temperature within the first twelve hours of infection. Between 1.5–2 dpi this initial febrile period began to wane and return to normal temperatures before increasing again between 2.5–4 dpi. The second febrile period had considerably more variability between individual macaques, often lasting out to 7–8 dpi and for several macaques there was more than one peak seen in that period. Two macaques infected with INH 9813, M5-19 and M6-19, were euthanized at 6 dpi to assess viral penetration into the CNS during the second febrile period. A significant decline in temperature (hypothermia) was observed in the post-febrile period in four of the macaques (Fig 1; 122–16, 165–16, 107–18, and 112–18).

We evaluated how inhaled dose impacted the febrile response to infection (Fig 2). For the overall post-exposure period, fever-hours (a measure of fever duration and severity) did not correlate with the inhaled dose ($r^2$ = 0.1593, p = 0.1574; Fig 2A). The febrile response was further broken down into two periods based on phase, the first period between 0.5–2 dpi and the second between 2.5–8 dpi. Fever-hours during the first febrile period did correlate with inhaled dose ($r^2$ = 0.5654, p = 0.00919; Fig 2B) while fever-hours in the second febrile period did not ($r^2$ = 0.0437, p = 0.4729; Fig 2C). We also calculated the maximum (max ΔT; Fig 2D–2F) and median (ΔT; Fig 2G–2I) deviations from the predicted temperature for each animal. Both max ΔT and median ΔT correlated well with inhaled dose during the overall post-exposure ($r^2$ = 0.4517 and 0.4686, respectively for max ΔT and median ΔT) and first febrile period ($r^2$ = 0.4130 and 0.5032 respectively for max ΔT and median ΔT). Max ΔT ($r^2$ = 0.3255) and median ΔT ($r^2$ = 0.3116) also correlated with inhaled dose during the second febrile period but not as strongly as the first febrile period or overall post-exposure period. There were no notable differences in the febrile response after TrD infection compared to INH-9813 but the TrD sample size was limited. We also evaluated whether there were differences in the febrile response based on sex or age. No significant differences were seen as a result of age other than the initial body weight of the macaque. The data suggests that female macaques might have somewhat lower febrile responses however this is confounded by dose and weight (females were smaller and exposed to lower doses). Further studies to eliminate these confounding effects would be necessary to verify differences based on sex.

## Clinical signs of neurological disease

Macaques were also observed daily after exposure for changes suggestive of neurological disease (Fig 3A). Overall, neurological signs were mild but prolonged in macaques infected with either strain of VEEV, with average scores between 2–3 after exposure continuing out to 20 dpi. Fig 3B shows reported signs suggestive of neurological disease and the frequency that these were observed in infected macaques. The most commonly observed signs were photophobia, mild tremors, and intermittent twitching (Fig 3B). The least frequently reported were those commonly seen in severe EEEV infection, such as ataxia and nystagmus. Seizures were not reported in any VEEV-infected macaques in this study. There were no significant differences in clinical signs between macaques infected with TrD or INH-9813.

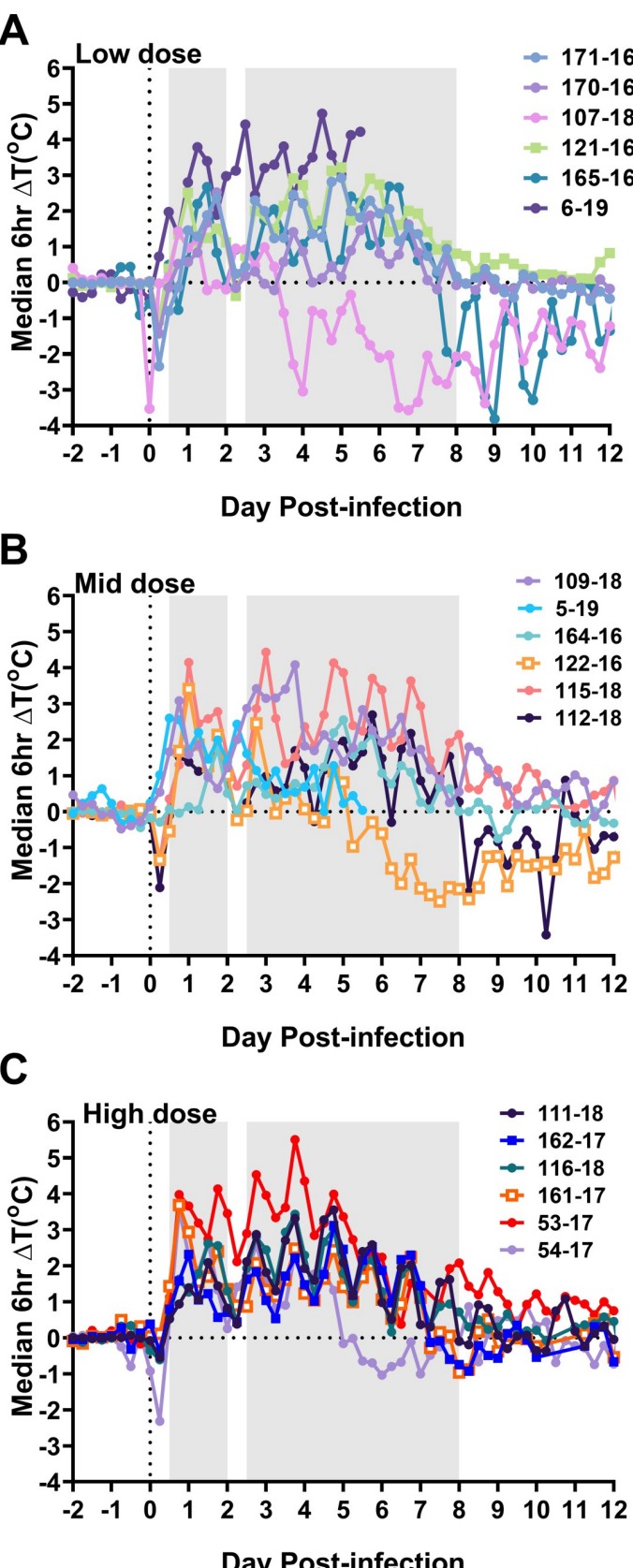

**Fig 1. Biphasic fever is a characteristic of VEEV infection in cynomolgus macaques.** Data from 16 VEEV infected NHPs are shown. For visual purposes, macaques were separated into 3 groups of 6 based on exposure dose. (A) low dose; (B) mid-dose; (C) high dose. Continuous fever data was collected and each data point represents six-hour median residual differences in actual versus predicted body temperature beginning four days prior to infection and continuing out to 12 dpi. Shaded areas on each graph represents the first (0.5–2 dpi) and second (2.5–8 dpi) febrile periods as determined from this data and prior studies. Open squares represent the animals exposed to TrD; all others were exposed to INH9813. Two animals, M5-19 and M6-19, underwent scheduled euthanasia at 6 dpi.

## Detection of CNS disease by telemetry

In addition to direct observation of neurological disease, a subset of VEEV-infected macaques were implanted with telemetry devices to monitor for changes in electroencephalography (EEG) and intracranial pressure (ICP). Group averages were taken for the VEEV-infected cohort of macaques and compared against baseline by repeated measures ANOVA. Fig 4 shows differences in 4 EEG wavebands, δ (delta; corresponding to deep sleep); θ (theta, corresponding to REM sleep); α (alpha; corresponding to a normal, wakeful state); and β (corresponding to periods of high activity) during each of the periods as denoted by infection and fever. While there was no significant change across all wavebands in the EEG power spectrum during the pre-symptomatic (0–1 dpi) or first febrile (1–2 dpi) periods, during the second febrile period (3–8 dpi), there was a significant decrease in all wavebands, denoted by asterisk-marked segments (Fig 4D). Although certain exceptions apply, such as small segments of discrete frequencies from 10–11 Hz and 22.5–26.75 Hz, all defined wave bands (δ, θ, α, and β) experienced an approximate 5% decrease from baseline ($p < 0.05$).

Substantial variation between macaques with regard to this finding occurs; indeed, significant individual macaque outliers in the pre-symptomatic period and the first febrile period (Fig 4B and 4C) precluded statistical significance, though trends of increase from baseline in certain wave bands such as in the δ and α wave bands during the first febrile period are visually apparent. In contrast, the downward change in all bands of EEG activity that occurs in concordance over the second febrile period, was statistically significant despite a relatively small effect size. The implications of a global decline in EEG activity in the second febrile period of VEEV suggest pathological neurological activity consistent with diffuse cerebral dysrhythmias and generalized slowing in EEG activity reported in human patients [34,35]. In the recovery period (Fig 4E), the powerbands returned to normal, though outlier individuals also exist. Circadian index, a subtraction of β from δ, also was significantly decreased during the second febrile period for VEEV-infected macaques relative to mock-infected macaques or baseline data (Fig 5). This suggests significant sleep disruption in the second febrile period. In the recovery period, circadian index returned to normal. In addition to the EEG changes described above, an increase in ICP was observed beginning 6 dpi and was significantly elevated on days 7, 8, and 9 (Fig 6). In a prior report we reported that in mock-infected macaques there were no changes in ICP after exposure [26].

## Hematological changes after VEEV infection

As has been reported previously with VEEV infection of non-human primates [9,10], after aerosol exposure to INH-9813 there was a significant drop in white blood cells in the first five days after infection, and most of that loss was in the lymphocyte population (Fig 7A). Granulocyte counts also dropped significantly in this time frame, while monocyte numbers increased. A significant increase in mean platelet volume was observed, which suggests mature platelets were being replaced with immature platelets at a higher rate than normal. Monocyte numbers increased notably during the second febrile period, peaking on day 6 post-infection. By 10 dpi, lymphocyte counts had increased to nearly double baseline counts while other leukocyte

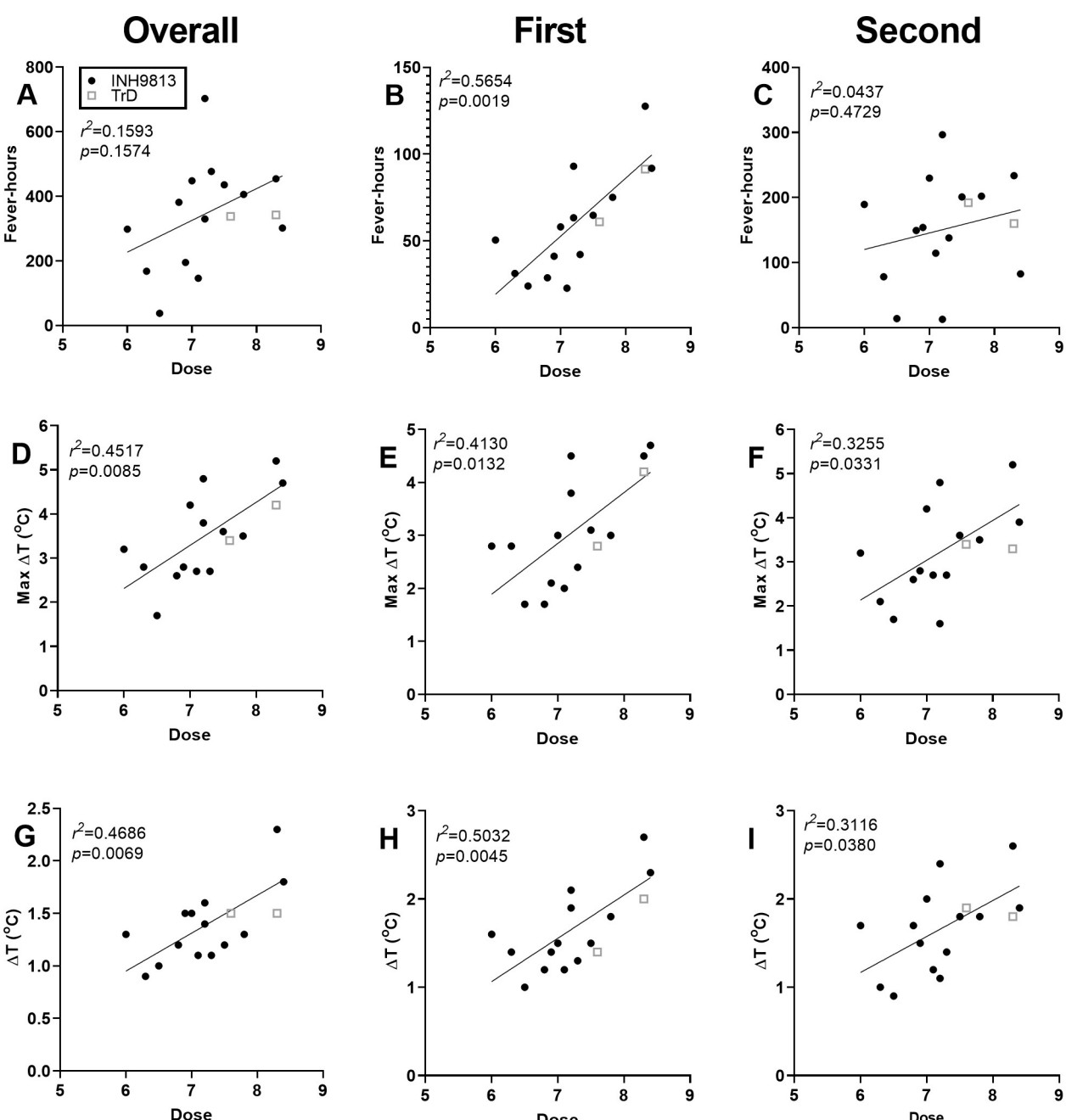

**Fig 2. Correlation between dose and fever is strongest for the first febrile period.** (A, B, C) Fever-hours, the sum of the significant elevations divided by four to convert to hours. (D, E, F) Maximum temperature deviation from pre-infection baseline for each individual animal. (G, H, I) Average temperature elevation above baseline.(A, D, G) Overall post-exposure period, (B, E, H) first febrile period, 0.5–2 dpi, (C, F, I) second febrile period, 2.5–8 dpi.

populations had returned to within normal ranges. By the study endpoint, all leukocytes were within normal ranges. Clinical chemistries showed a significant drop in albumin, amylase, and total protein between 3–5 dpi, during the second febrile period, although the most profound changes were in albumin (Fig 7B). By 10 dpi these values returned to pre-infection levels. Other parameters evaluated (BUN, ALT, CRE) did not significantly change post-infection.

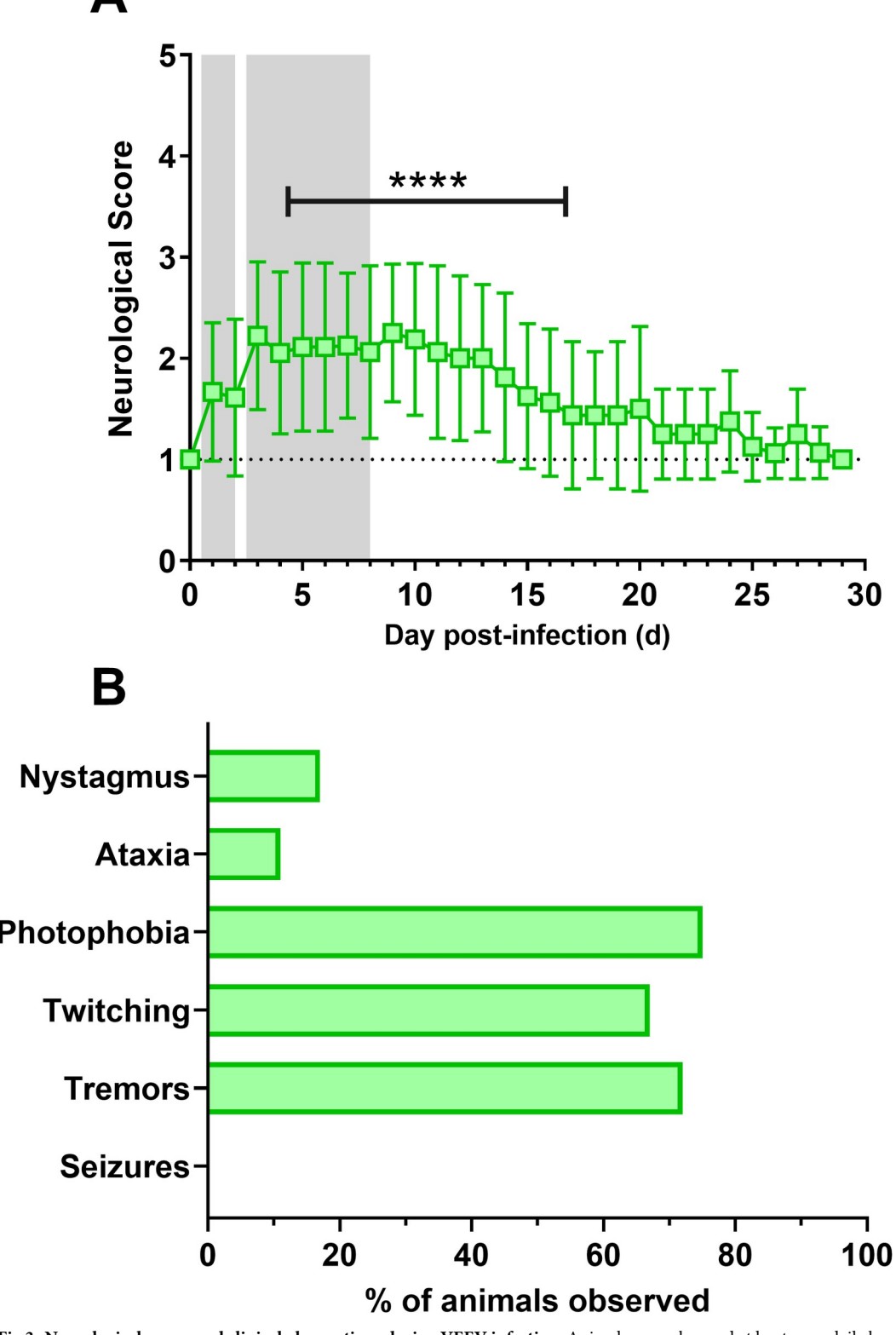

**Fig 3. Neurological scores and clinical observations during VEEV infection.** Animals were observed at least once daily by research staff, scored according to IACUC-approved parameters, and clinical observations were recorded. (A) daily neurological scores (+/- SD). Asterisks indicate scores that are significantly above pre-infection levels based on one-way ANOVA analysis with multiple comparisons. (B) frequency of clinical observations from (n = 18).

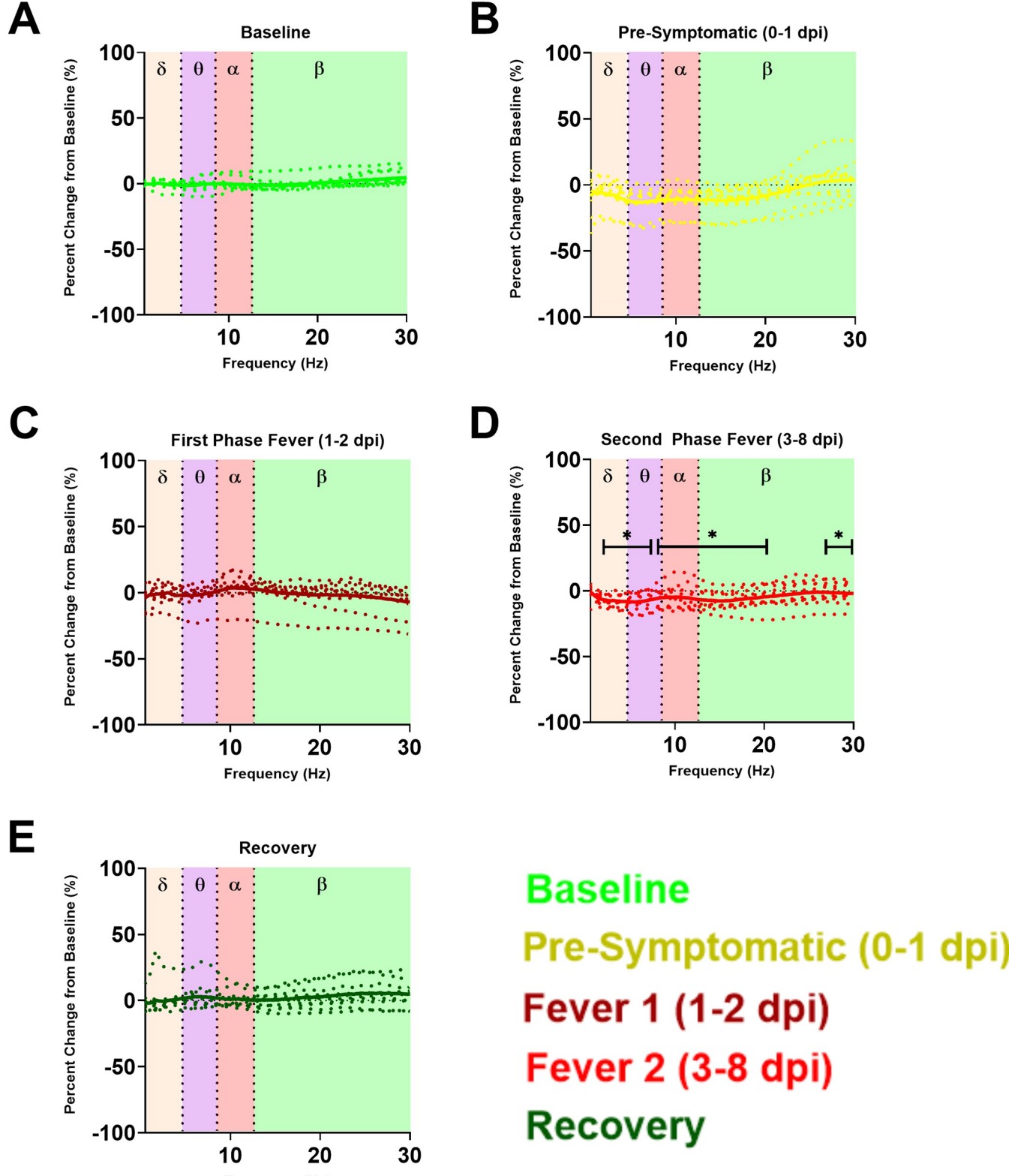

**Fig 4. EEG power spectral density plot magnitude as percentage change from baseline in VEE.** Power spectral density plotted as percent change from A) baseline in cohorts of VEEV-infected macaques from B) pre-symptomatic, C) first phase fever, D) second phase fever, and E) recovery. Solid traces indicate group average percentage change while dotted traces illustrate percentage change in individual macaques. Wave bands are color coded by shaded boxes; the

box plots at each discrete frequency segment are compared between baseline, pre-symptomatic, febrile phases of the biphasic fever, and recovery periods. Repeated measures ANOVA and post hoc pairwise comparisons at each discrete frequency showed a statistically significant decrease in all wavebands during the second phase of the biphasic fever ($p<0.05$).

### Viremia

Transient viremia is also a hallmark of VEEV disease in humans and has been reported in macaques after aerosol exposure [9,10]. By qRT-PCR, virus was detected in the blood of macaques after aerosol exposure to INH-9813 on days 2 and 4 post-infection at levels ranging from 1–4 $\log_{10}$ pfu while by plaque assay virus was only found in the blood on day 2 (Fig 8A and 8B). This is similar to what has been reported for viremia in nonhuman primates after aerosol exposure to other VEEV isolates [9,10,13]. Nasal swabs were positive at low levels for virus by qRT-PCR at 2 and 4 dpi and in one macaque out to 8 dpi. However, no infectious virus was detected by plaque assay from nasal swabs. This is in contrast to reports of throat swab titers with prior VEEV studies in macaques [13], however, it may be a result of the difference in the site being sampled (nasal vs throat) or the virus used.

### Long-term presence of VEEV RNA in the CNS

CNS samples collected at necropsy were assessed for the presence of VEEV RNA by qRT-PCR (Fig 9). For one of the two macaques euthanized on day 6, virus was found by qPCR in all CNS

**Fig 5. Circadian index power ratios suggest circadian disruption during second febrile phase of VEE.** Repeated measures ANOVA of circadian index power ratios of mock-infected macaques and macaques infected with VEEV show the suppression of circadian variation during the second febrile peak compared to baseline between VEEV-infected and mock-infected macaques ($p = 0.0023$).

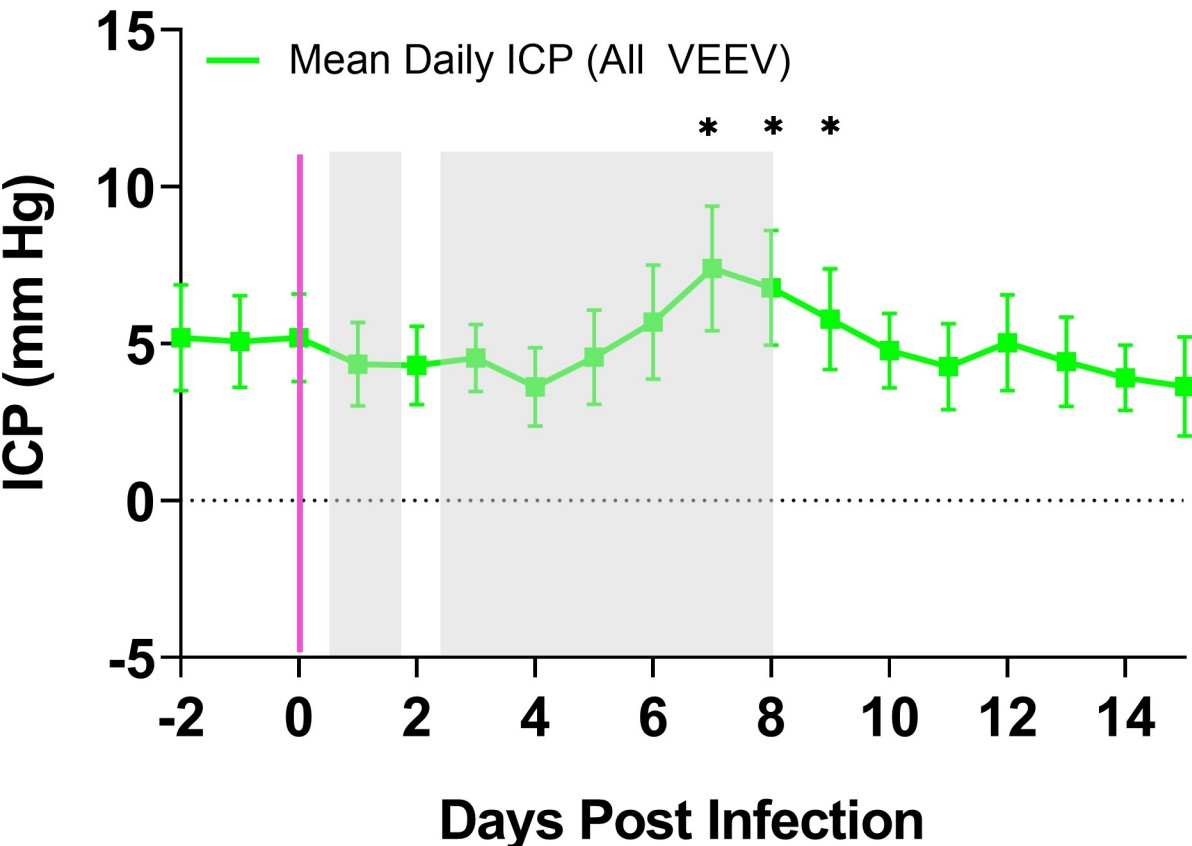

**Fig 6. Intracranial Pressure in VEEV-infected NHPs.** Mean ICP (daily mean +/- SEM) in VEEV-infected macaques (n = 6) superimposed upon gray boxes indicating febrile period. Magenta line indicates aerosol challenge. Two-factor repeated measures ANOVA with post-hoc pairwise comparisons found a significant increase in ICP compared to mock-infected macaques on days 7–9 post infection, denoted by the asterisk-topped bracket.

tissues examined except CSF. In the other animal, however, virus was only found in the cervical lymph node. For macaques that survived VEEV infection (euthanized between 26–52 dpi), no infectious virus was found at necropsy by plaque assay. VEEV RNA was found in the CNS of macaques euthanized on 26–35 dpi, with titers ranging from 1–4 $\log_{10}$ vRNA copies/g of tissue including the frontal and parietal lobes (7 of 10 macaques), temporal lobes (8 of 10), occipital lobes (6 of 10) and the cerebellum (5 of 10). In addition, cervical lymph nodes (CLN) were positive for VEEV RNA in 12 of 14 macaques euthanized between 26–52 dpi. Cerebrospinal fluid (CSF) was also positive in 9 of 14 macaques euthanized at 26–52 dpi, including 3 macaques euthanized between 48–52 dpi. These results indicate that VEEV vRNA can remain detectable in the brain for at least 4–5 weeks after infection, and longer in the CSF and CLN, under these conditions.

## Inflammatory and traumatic brain injury response in the CNS

Expression of inflammatory cytokines and chemokines in the CNS after VEEV infection was evaluated by multiplex analysis (Fig 10). Two chemokines, MCP-1 and IP-10, were significantly elevated in VEEV-infected animals both at 6 dpi and at necropsy (28–35 dpi). For

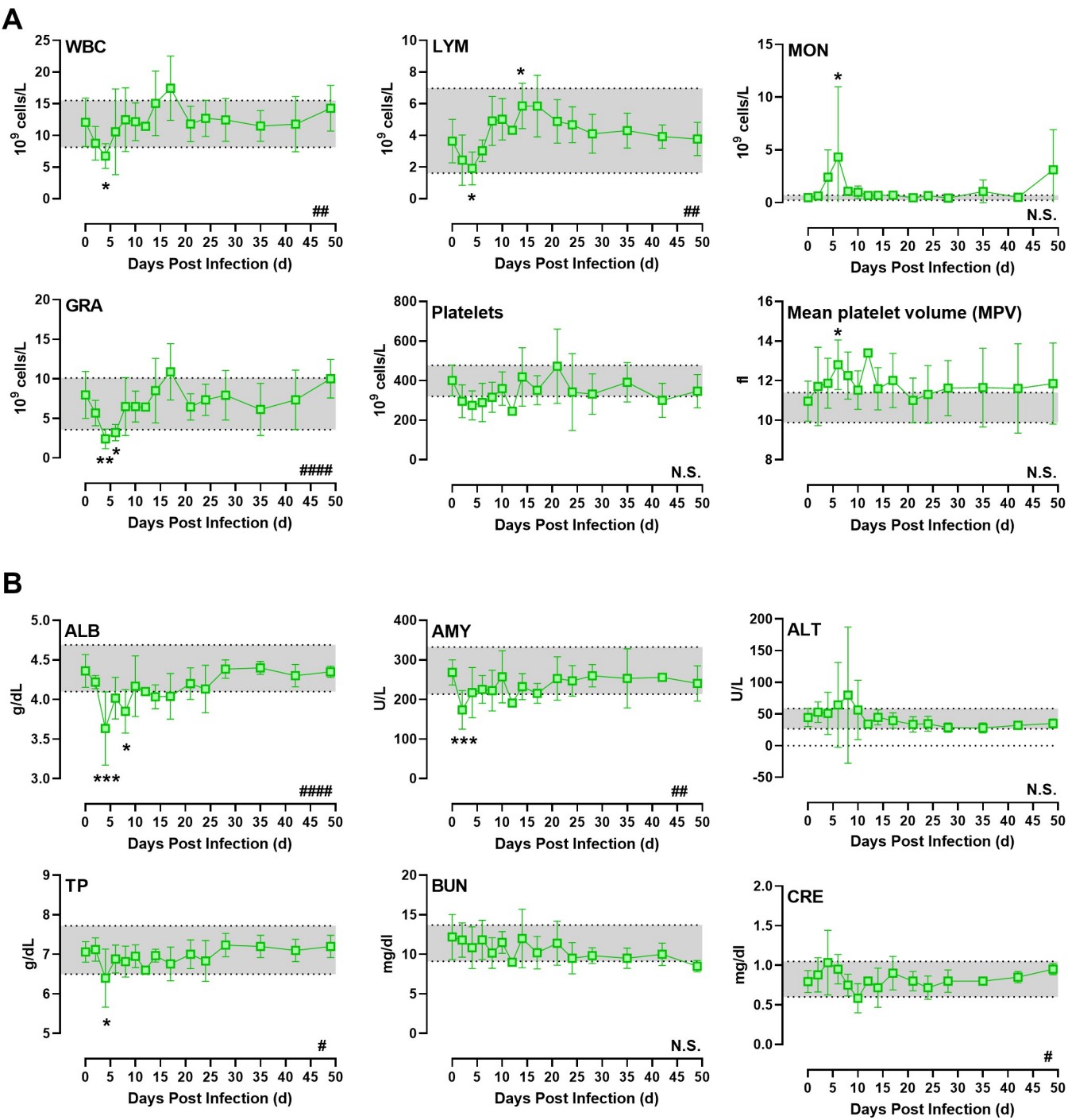

**Fig 7. Alterations in CBC and blood chemistry parameters in VEEV-infected NHPs.** Blood was sampled every 2 days and the average CBC results from VEEV-infected NHPs are shown (n = 6). Shaded area represents the range (min-max) of the mock-infected control animals. Data shown is up to 48 dpi. Hashtags in lower right corner of each graph indicate significance determined by 2-way ANOVA. Asterisks above or below data points indicate significance compared to 0 time point determined by multiple comparisons within ANOVA.

comparison, similar samples were analyzed from 4 mock-infected macaques. In all surviving macaques as well as the two euthanized at 6 dpi, elevated levels of MCP-1 were found consistently in the CSF and sporadically throughout the CNS tissue (Fig 10A). IP-10 levels were also

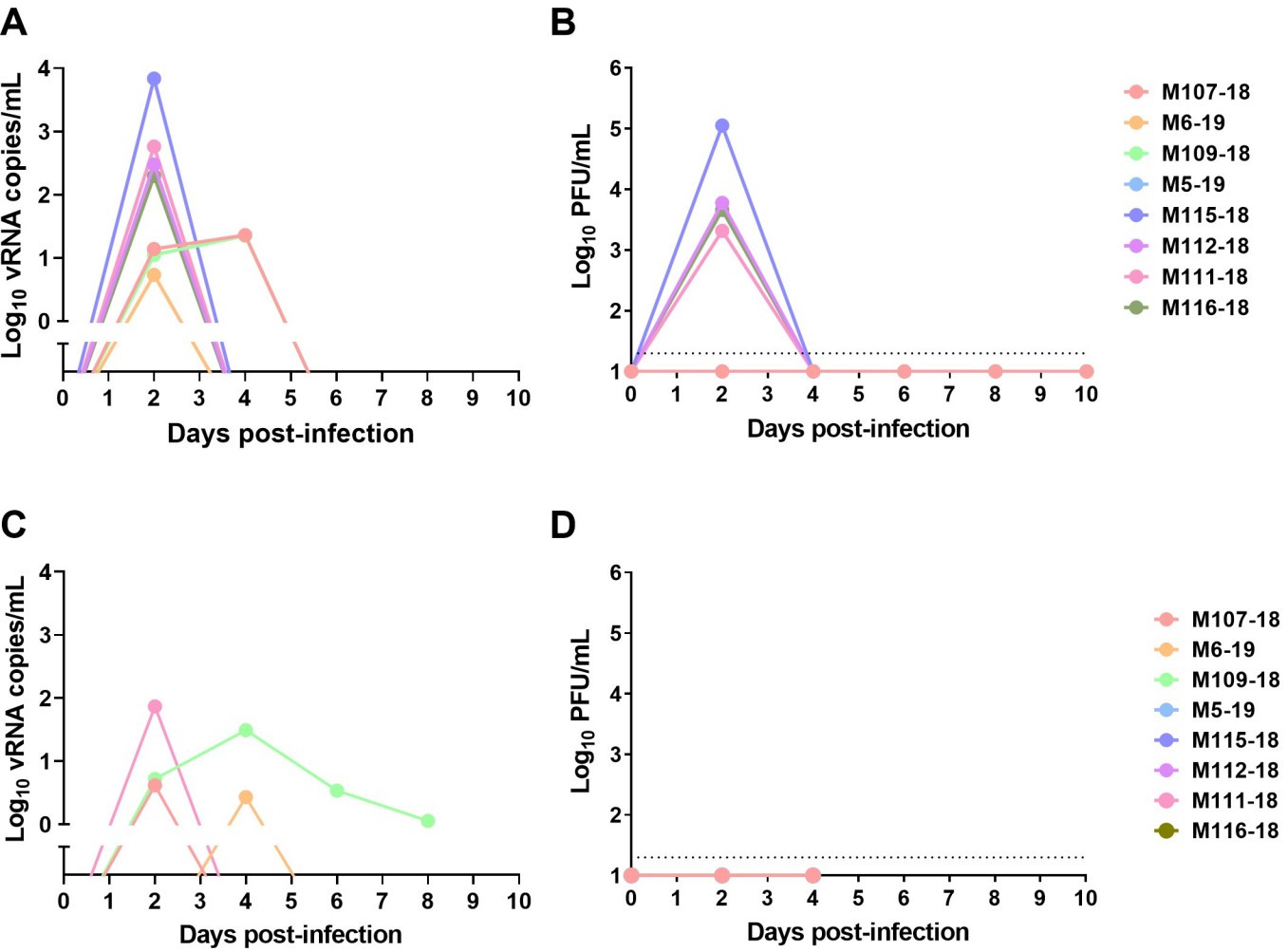

**Fig 8. Longitudinal plasma viremia and nasal swabs during VEEV infection.** Longitudinal plasma and nasal swabs were taken every 2 days after infection. vRNA was measured by semi-quantitative q-RT-PCR and infectious virus by plaque assay. (A) plasma vRNA; (B) plasma viremia; (C) nasal swab vRNA; (D) nasal swab plaque assay. $Log_{10}$ vRNA titers in A and C have background subtracted. LOD of plaque assay is indicated by horizontal dashed line in B and D.

elevated in the CSF and more sporadically in the brain tissues (Fig 10B). Animals with consistently elevated MCP-1 and IP-10 levels, such as M122-16, M107-18, and M109-18, also had more vRNA compared to 164–16 and 170–16 (Fig 9). Given the small number of macaques, the results were suggestive, not significant. IL-6, which we found was elevated in the brains of EEEV-infected macaques [36], was only elevated in one tissue from one surviving macaque and one of the two macaques at 6 dpi (Fig 10C).

We also recently reported significantly elevated levels in two of three markers of traumatic brain injury in the thalamus and cerebellum after lethal EEEV infection [36]. We examined the expression of these markers in VEEV infection and found significant elevations of MMP-9 and LIF in the thalamus but not the cerebellum of VEEV-infected macaques at 6 dpi (Fig 11). GFAP levels were unchanged compared to mock-infected macaques in either tissue. This may suggest there is some disruption of the blood-brain barrier on day 6 after infection with VEEV but it is not as severe as that seen with EEEV.

In NHP models of VEEV disease, characterization of leukocytes infiltrating the CNS had not previously been documented. We used our previously developed procedure for isolating

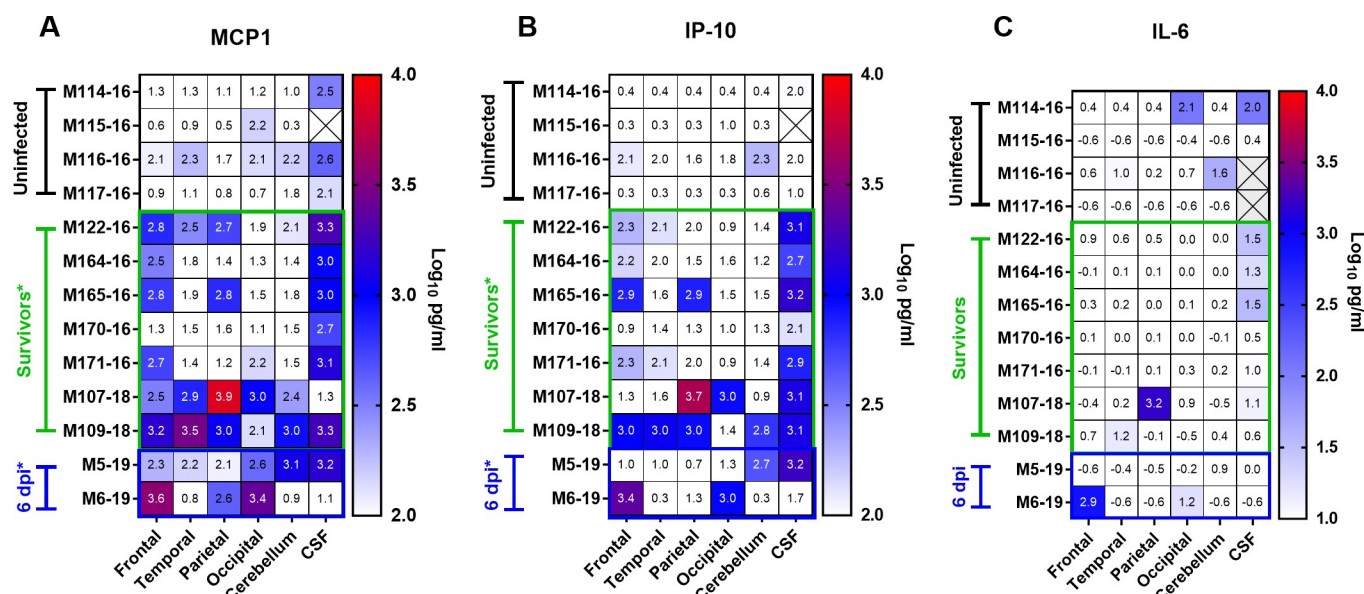

**Fig 9. VEEV RNA levels in the tissues of NHPs at necropsy.** (A) Brain tissue, cervical lymph nodes (CLN), and cerebrospinal fluid (CSF) from VEEV-infected animals euthanized at the end of the study were subjected to q-RT-PCR for viral RNA. Numbers in each cell represent the $\log_{10}$ vRNA copies/g or ml (for CSF) after subtraction of background and correspond to the heatmap colors. Blank/white cells were below the limit of detection (LOD). Infectious virus was not detectable in these samples by plaque assay. Numbers on x-axis indicate day post-infection (dpi) that the animals were euthanized. Two animals (M5-19 and M6-19) were euthanized at 6 dpi during acute disease; the rest were >25 dpi at the end of our study period. The cell with an X represents a tissue sample that was not available for testing.

**Fig 10. Cytokines in the CSF of VEEV-infected NHP at necropsy.** Multiplex analysis was used to measure cytokines in the CSF of mock-infected or surviving VEEV-infected animals at necropsy. (A) MCP-1. (B) IP-10. (C) IL-6. Statistical significance determined by 2-way ANOVA with multiple comparisons. In (A) and (B), both groups of VEEV-infected animals had elevated chemokine levels compared to mock infected controls.

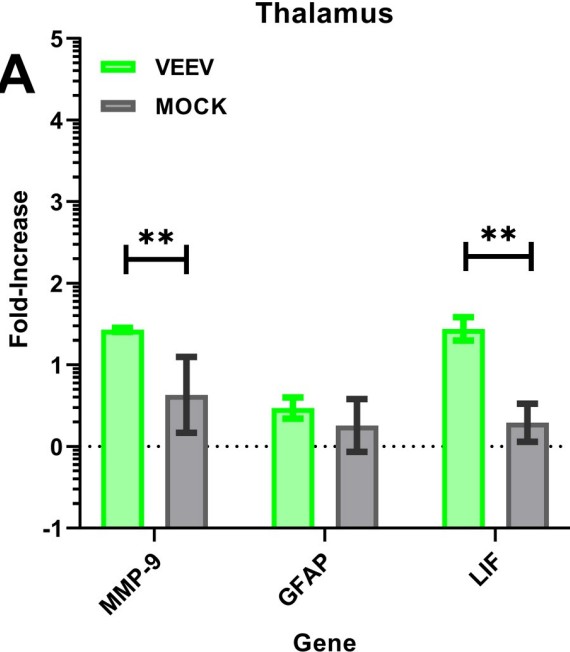

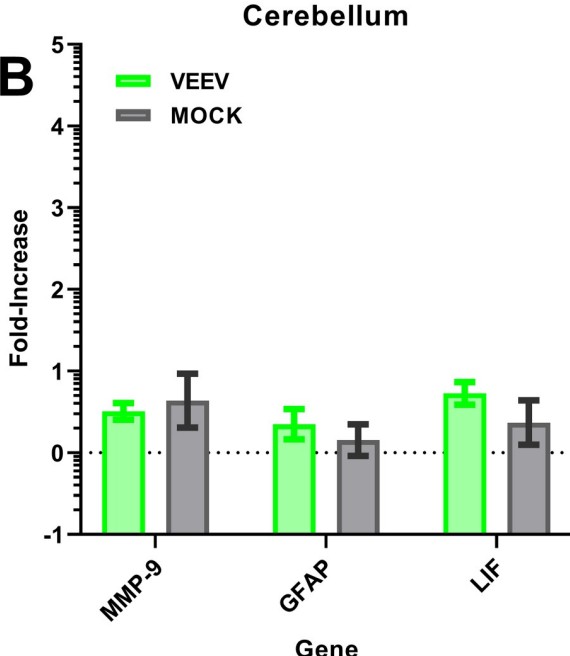

**Fig 11. Increase in traumatic brain injury gene expression in VEEV-infected macaque brains.** qRT-PCR was used to evaluate expression of mRNA for three genes associated with traumatic brain injury in the brains of VEEV-infected macaques euthanized on day 6. Similar analysis was also done on mRNA from the brains of mock-infected macaques. Two regions of the brain were evaluated, the A) thalamus and B) cerebellum. Fold-change is in comparison to GAPDH expression. Asterisks indicate where statistical analysis indicated a significant difference between VEEV-infected and mock-infected samples; * $p \leq 0.05$, ** $p \leq 0.01$, *** $p \leq 0.001$.

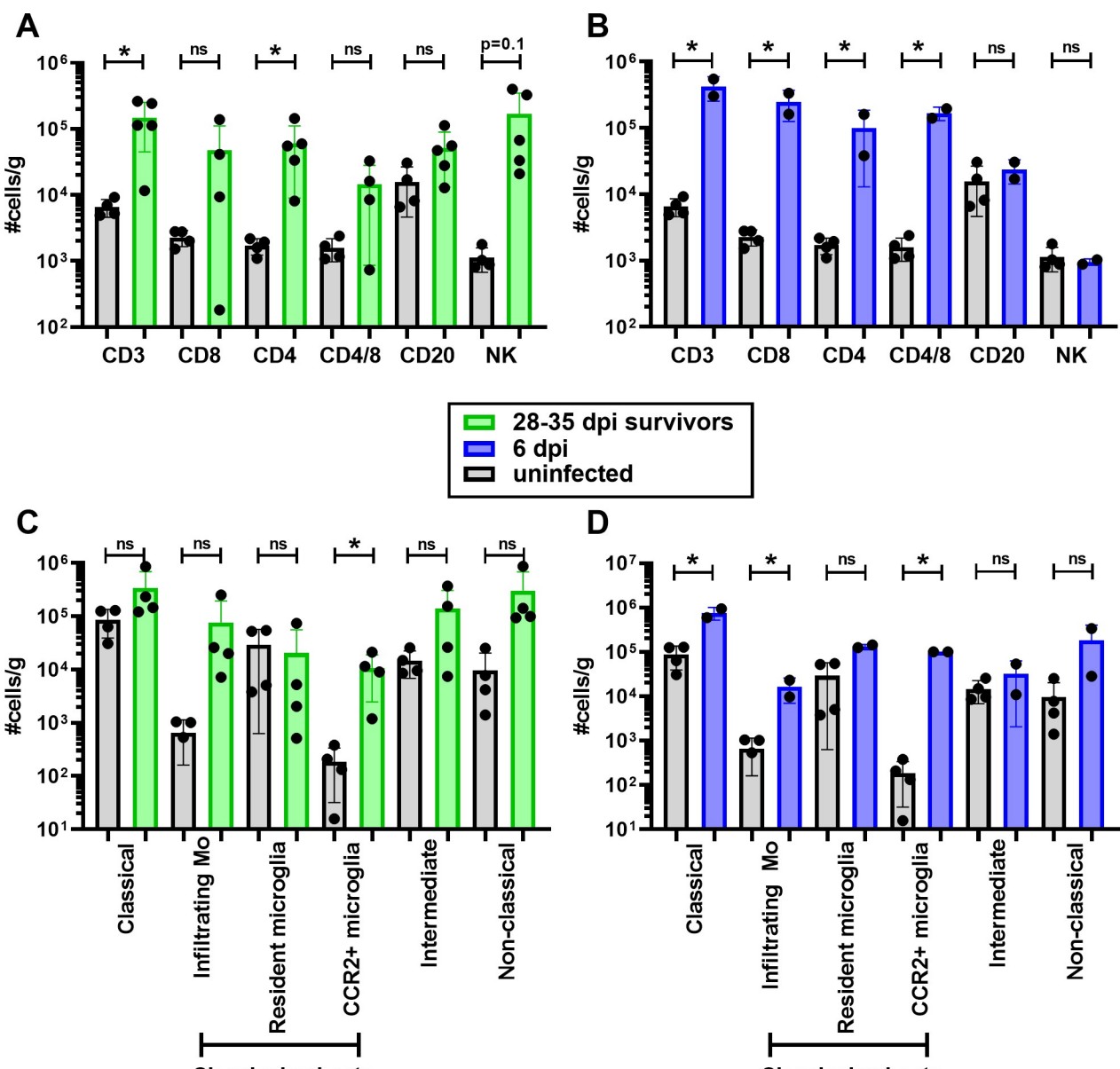

**Fig 12. Leukocyte infiltration into the brain of VEEV-infected macaques.** Leukocytes were isolated from brain tissue and phenotyped using flow cytometry (gating strategy from [36]). (A,B) Lymphoid and (C,D) myeloid lineage cells from mock-infected (gray; n = 4), survivors (green; n = 4; 28–35 dpi), and animals euthanized at 6 dpi (blue; n = 2). For each animal, cells were isolated from 5 brain regions (frontal, temporal, parietal, occipital, cerebellum) and each circle represents the average #cells/gram for each animal for all 5 regions. Bars indicate group averages +/- SD. Statistically significant differences were determined using 2-way ANOVA with multiple comparisons. Significance between mock-infected and infected animals is indicated by the asterisk above each group.

leukocytes from CNS tissues and gating strategy for identification of lymphoid and myeloid populations using flow cytometry [36]. For each animal, leukocytes were isolated from 5 brain regions (frontal, temporal, parietal, occipital, cerebellum) and averaged to obtain the data points in Fig 12. In surviving macaques, CD3+ T cells remained significantly elevated even at >28 dpi (Fig 12A). Of the CD3+ T cells, CD4+ cells were significantly increased in all 5 VEEV-infected macaques examined while CD8+ cell numbers were elevated in only 3 of 5 macaques and CD8+ cell increases were not significantly different from mock-infected. NK

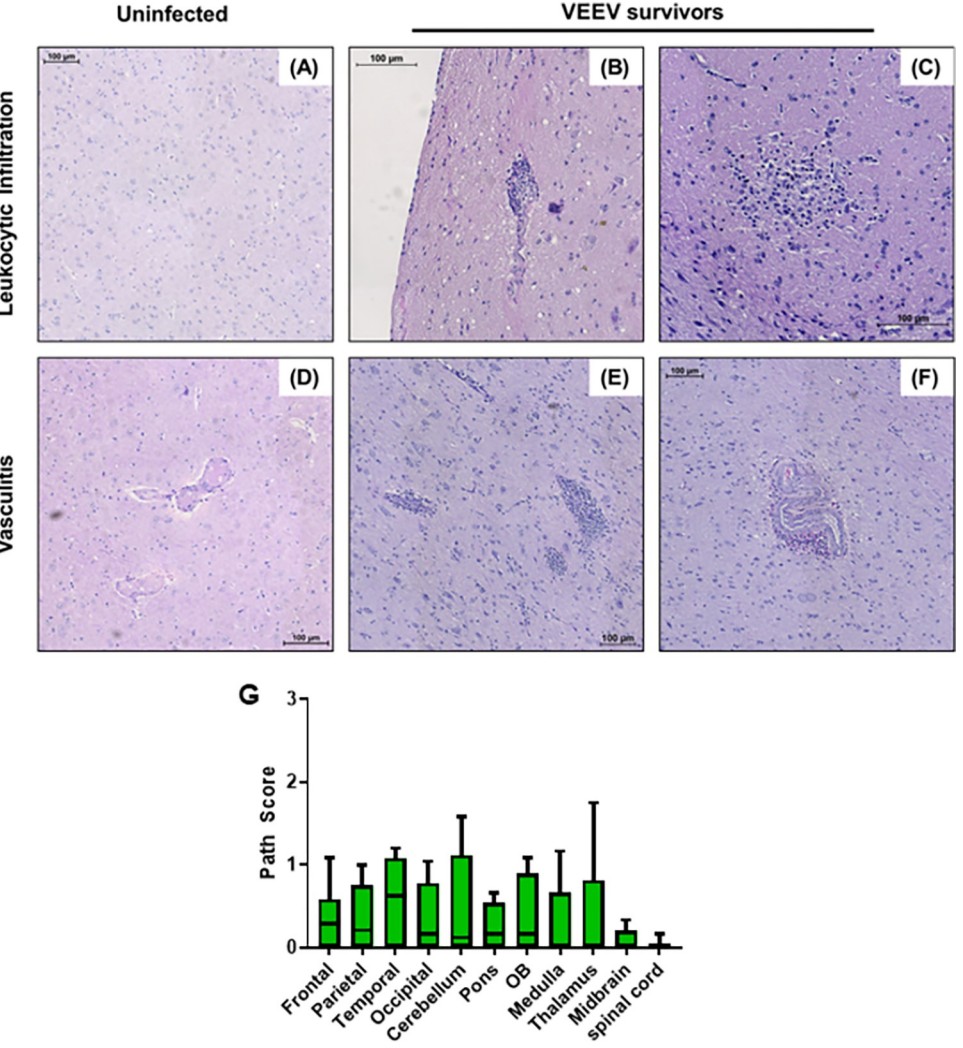

**Fig 13. Histopathologic lesions in the thalamus of non-human primates (NHPs) infected with Venezuelan Equine Encephalitis Virus (VEEV).** (A) Thalamus control: M117-16. (B) Thalamus infected, focal leukocytic infiltrate: M164-16. (C) Thalamus infected, diffuse leukocytic infiltrate: M121-16. (D) Thalamus control, M117-16. (E, F) Thalamus infected, perivascular leukocyte infiltration: M121:16. (G) Formalin-fixed tissue samples from 6 NHP were subjected to H&E staining and scored 0–4 based on severity of lesions.

cell numbers were substantially elevated but not significantly different compared to mocks (p = 0.1). In animals sacrificed at 6 dpi, there were significantly elevated numbers of T cells from both the CD4+ and CD8+ lineages, but B cells and NK cells were not elevated (Fig 12B). Fewer changes were seen overall in myeloid lineage cells, but both classical and infiltrating macrophage counts were significantly elevated at 6 dpi, as well as CCR2+ microglia. In survivors, only CCR2+ microglia counts were significantly different from controls, although there was a trend with both infiltrating monocytes and non-classical monocytes.

## Pathological changes in the CNS

CNS tissues recovered from VEEV-infected macaques euthanized at 26–35 dpi were also evaluated for pathological changes (Fig 13). Although these macaques had fully recovered from the infection according to physiological responses (temperature, EEG, ICP as well as clinical

observations), there remained evidence of mild, diffuse inflammation throughout CNS regions of some VEEV-infected macaques (Fig 13G), which occurred concomitant with the elevated leukocytes (Fig 12), cytokines (Fig 10), and the continuing presence of vRNA (Fig 9).

## Discussion

We present here data from studies conducted to advance the cynomolgus macaque model of VEEV using an infectious clone-derived stock of a human isolate of VEEV (INH-9813) that has been selected for use in FDA Animal Rule efficacy studies [12]. Previous studies have shown that aerosol exposure to VEEV resulted in a biphasic fever, lymphopenia, viremia, and mild clinical signs indicative of neurological disease [9,10]; however, all of these studies utilized cell passaged biological isolates of uncharacterized genotype, which is not optimal for licensure of drugs and biologics using the FDA's Animal Rule. Despite the strain limitations, these previous studies have been useful in evaluating potential VEEV vaccine candidates [13,14,37,38]. Previous studies also did not perform detailed analyses of the physiological and pathogenic changes in macaques using a clone-derived human strain of VEEV, which we document here. Clinical signs of neurological disease in macaques reported previously were mild (tremors, loss of coordination, anorexia, dehydration), calling into question whether they truly represented CNS disease or simply febrile illness. Here, we have made use of more advanced radiotelemetry, measuring changes in the brain by EEG and ICP in concert with body temperature and clinical observations to better ascertain the neurological effects of VEEV infection in macaques. This has aided in determination of potential biomarkers suitable for therapeutic intervention studies as well as provided new insights into the disease caused by VEEV and relevance of the macaque model to human disease. A notable finding in this regard is the presence of viral RNA, inflammatory cells and cytokines in the CNS and lymph nodes up to 50 days post infection, which suggests that VEEV replication may be persistent and provides a new target for assessment of therapeutic efficacy. Overall, the data presented here add further evidence to the utility of the cynomolgus macaque as a model suitable for evaluating drugs or biologics (vaccines) for efficacy against aerosol exposure to VEEV.

Prior work with VEEV has predominantly used cell-culture passaged VEEV isolates [12]. The prototypical epizootic VEEV isolate used in many studies was TrD, which was original isolated from a donkey in 1943. However, recent guidance from the FDA on the licensure of drugs and biologics using the FDA's "Animal Rule" has stated that the FDA would prefer the use of well-characterized, low-passage isolates of pathogens from human patients. In a 2019 workshop, an IC serotype isolate (INH-9813) was selected for pivotal efficacy studies that would support licensure using the Animal Rule [12]. Use of infectious clones of low-passage isolates would prevent concerns about cell-culture adaptations resulting in attenuation [21,39]. We demonstrated here that the febrile response and disease in macaques after aerosol exposure to an infectious clone of INH-9813 was comparable to that seen with virus derived from an infectious clone of TrD. These results also compare well with previously published data for other infectious clones and biological virus isolates [9,10]. The data presented here do not suggest any impact on disease course as a result of the age of the macaque although it should be noted that none of the macaques were elderly and the youngest was a juvenile adult. The data suggest that the disease course in female macaques might be less disease than what is seen in males but there were differences in the dose and weight of the which confound this analysis. Additional study will be necessary to study whether there are any age- or sex-related changes in macaques infected with VEEV.

A hallmark of VEEV disease is the biphasic fever seen in humans. Macaques have a similar biphasic fever after aerosol exposure to VEEV, as shown previously and in this study [9,10]. In

natural VEEV infection, there is initially peripheral replication of the virus, predominantly in lymphoid organs, followed by spread to the CNS. Clinical signs indicative of CNS disease are more prominent in the second febrile period, suggesting that viral penetration & replication in the brain more likely occurs during this period in both humans and macaques [9,34]. The EEG and ICP data we have reported here are consistent with this notion. In the two macaques euthanized at 6 dpi, one had VEEV RNA in all CNS tissues tested while the second did not but had virus in the cervical lymph node sample examined. This compares well with the febrile response in these animals, as the macaque with virus in the CNS had a strong febrile response (+4°C) prior to euthanasia at 6 dpi while the macaque with no virus in the CNS had <1°C elevation in temperature at 5–6 dpi.

Analysis of fever data collected by telemetry has been shown to be very useful in assessing vaccine efficacy in cynomolgus macaques, particularly for VEEV [10,13,14,37,38]. Therapeutic studies would require a biomarker that would initiate treatment and there was a question whether fever was specific enough to serve as that marker since cynomolgus macaque studies to date have not assessed when VEEV reaches the CNS. One of the goals of this study was to ascertain whether EEG and ICP telemetry data might be suitable as that trigger. Given the mildness of the observed neurological signs with VEEV compared to EEEV, which is typically lethal, it was not clear prior to these studies if EEG and ICP data collected from telemetry would be useful in determining CNS effects of infection. However, the data showed a significant decrease in overall EEG power bands in the second febrile period followed by a significant increase in ICP late in the second febrile period. Analysis of EEG and ICP telemetry data requires considerable time and effort, and cannot be visually ascertained while data is being collected 'live'. Further, the late rise in ICP in relation to the second febrile period and neurological signs observed would suggest that an increase in ICP seen by telemetry lags behind CNS disease. Based on this, at least at this time, such data would not be useful as a trigger to initiate treatment. However, considering that the differences were significant to mock-infected macaques, this data could be useful in ascertaining the efficacy of vaccines or therapeutics in preventing CNS disease caused by VEEV. In our opinion, the data suggests that fever is suitable as a trigger, however, longitudinal studies of VEEV disease in cynomolgus macaques to include when and how the virus reaches the CNS will be critical in making this determination.

Laboratory findings reported here are also in keeping with prior reports of VEEV infection in cynomolgus macaques, which included lymphopenia, mild changes in clinical chemistries, and a low, transient viremia in the blood during the first 5 dpi. Somewhat surprisingly, no infectious virus was found in nasal swabs after infection although viral RNA was found. Previous studies have found virus in the throat of macaques infected with VEEV via aerosol or subcutaneous inoculation [9,13]. One possible explanation for the lack of infectious virus in the nasal regions here is that there may be regional differences in viral replication. Throat swabs would include the tonsillar region, which might support high levels of viral replication early after infection given the strong myeloid cell tropism of VEEV [30]. Alternatively, it is possible that differences in the virus isolates used might explain the differences in mucosal virus isolation.

The most surprising findings in this study were the presence of inflammatory cytokines and elevated T cells as well as VEEV RNA in the CNS of macaques that appeared to have recovered fully from the disease. Both IP-10 and MCP-1 have been shown to serve as chemokines for leukocytes, particularly T cells and monocytes/macrophages, into tissues. The levels of these chemokines, as well as the number of T cells and macrophages found in the CNS, at the day 6 time point relative to study endpoint would suggest the inflammatory response is resolving at study endpoint. It is notable that the gene expression of the traumatic brain injury markers MMP-9 and LIF on day 6 is only elevated in the thalamus while GFAP was not

elevated; this stands in contrast to our previous study with eastern equine encephalitis virus (EEEV) where all three markers were elevated (and to a greater extent) in both thalamus and cerebellum on day 6 [36]. This suggests the inflammatory response to VEEV is more limited than the response to EEEV but further investigation is needed to explore this response and how long it persists.

Viral RNA was found throughout the brain out to 35 dpi and in the CSF and CLN out to 52 dpi in 3 of 4 macaques. The number of animals, the level of virus found, and the failure to amplify virus in mock-infected macaques suggests this finding is not artifactual. No infectious virus was seen by plaque assay; however, it is possible that antibody present at that point may have neutralized free virions. Consistent with the finding of viral RNA in the CNS past 28 dpi, MCP-1 and IP-10 levels remained elevated in the CSF of all surviving VEEV-infected macaques examined and sporadically in other CNS tissues. Furthermore, the increased numbers of T cells and CCR2+ microglia in the CNS after 28 dpi as well as the histopathological lesions found in the CNS out to 52 dpi suggest an ongoing inflammatory process. Considering that long-term sequelae have been reported in human cases [40], these data suggest the macaque will be useful for investigating this aspect of VEEV disease and that viral persistence may be a useful marker for evaluation of therapeutic and vaccine efficacy in this model. Further study is needed to determine whether or not this viral RNA represents intact, replicating virus, the duration of viral persistence in the host, the characteristics of resolution of the inflammatory response and their relationship to pathological changes in the CNS.

## Supporting information

**S1 Table. Summary of monkeys, challenge doses and fever responses.** a Virus = isolate used; Age: years; Weight: kg; Dose: $\log_{10}$ pfu. b Rmax = maximum residual difference between actual and predicted temperature. c Duration: hours. d Fever-hours: sum of significant temperature elevations, divided by four to convert to hours. e Ave Elev: average elevation, fever-hours / duration. f Overall = fever for the entire post-challenge period. g First = fever during the first fever period, 0.5–2 dpi.
(DOCX)

**S2 Table. PCR primers used in amplifying viral RNA for sequencing.**
(DOCX)

**S1 Data. Virus sequence obtained by Sanger sequencing.**
(PDF)

## Acknowledgments

Any opinions, findings, and conclusions or recommendations expressed in this material are those of the author(s) and do not necessarily reflect the position or the policy of the government and no official endorsement should be inferred.

## Author Contributions

**Conceptualization:** William B. Klimstra, Amy L. Hartman, Douglas S. Reed.

**Data curation:** Henry Ma, Joseph R. Albe, Theron Gilliland, Ivona Pandrea, Tobias Teichert, Stacey Barrick, Amy L. Hartman.

**Formal analysis:** Henry Ma, Cynthia M. McMillen, Ivona Pandrea, William B. Klimstra, Amy L. Hartman, Douglas S. Reed.

**Funding acquisition:** William B. Klimstra, Amy L. Hartman.

**Investigation:** Joseph R. Albe, Theron Gilliland, Cynthia M. McMillen, Christina L. Gardner, Devin A. Boyles, Emily L. Cottle, Matthew D. Dunn, Jeneveve D. Lundy, Noah Salama, Katherine J. O'Malley, Tobias Teichert.

**Methodology:** Henry Ma, Joseph R. Albe, Theron Gilliland, Cynthia M. McMillen, Matthew D. Dunn, Tobias Teichert.

**Project administration:** Stacey Barrick, William B. Klimstra, Amy L. Hartman, Douglas S. Reed.

**Resources:** Stacey Barrick, William B. Klimstra, Amy L. Hartman.

**Software:** Henry Ma.

**Supervision:** Stacey Barrick, William B. Klimstra, Amy L. Hartman, Douglas S. Reed.

**Writing – original draft:** William B. Klimstra, Amy L. Hartman, Douglas S. Reed.

**Writing – review & editing:** Henry Ma, Joseph R. Albe, Cynthia M. McMillen, Christina L. Gardner, William B. Klimstra, Amy L. Hartman, Douglas S. Reed.

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
