## [Decision Letter · Decision Letter 0]

18 Oct 2021

Dear Dr. Reed,

Thank you very much for submitting your manuscript "Long-term persistence of viral RNA and inflammation in the CNS of macaques exposed to aerosolized Venezuelan equine encephalitis virus." for consideration at PLOS Pathogens. As with all papers reviewed by the journal, your manuscript was reviewed by members of the editorial board and by several independent reviewers. In light of the reviews (below this email), we would like to invite the resubmission of a significantly-revised version that takes into account the comments of the reviewers, all of whom have indicated interest in the work but who, among other concerns, see room for substantial improvement in presentation and description of background, data, and figures.

We cannot make any decision about publication until we have seen the revised manuscript and your response to the reviewers' comments. Your revised manuscript is also likely to be sent to reviewers for further evaluation.

Sincerely,

Jens H. Kuhn

Associate Editor

PLOS Pathogens

Mark Heise

Section Editor

PLOS Pathogens

Kasturi Haldar

Editor-in-Chief

PLOS Pathogens

orcid.org/0000-0001-5065-158X

Michael Malim

Editor-in-Chief

PLOS Pathogens

orcid.org/0000-0002-7699-2064

Reviewer's Responses to Questions

**Part I - Summary**

Reviewer #1: The manuscript describes the long-term persistence of vRNA and inflammation in the CNS of NHPs exposed to VEEV via the aerosol route of exposure. The challenge virus was VEEV INH-9813 that was generated via reverse genetics and was delivered via small particle aerosol. The paper addresses some key questions regarding the NHP model of VEEV, particularly regarding the more long-term sequelae associated with disease. Overall, the study is well done, however, it would benefit from including more experimental and analysis details that will enable future comparisons possible.

Reviewer #2: In this manuscript, the authors present their finding on small aerosol exposure of Cynomolgous macaques with Venezuelan equine encephalitis virus. While previous studies have been published on aerosol exposure of primates with VEEV, this study utilized clonally derived stocks of the INH-9813 strain selected for preclinical efficacy studies under the FDA animal rule. Additionally, this study employed radiotelemetry to track fever and measure electroencephalography and intracranial pressure providing greater insight into CNS disease caused by VEEV infection. Results from this analysis provided additional markers that could be used to assess drug/vaccine efficacy. Lastly, the authors report that VEEV viral RNA along with pro-inflammatory cytokines/chemokines and leukocytes are detectable in CNS tissues associated with pathological changes at 50 or more days post infection. This finding is of significant importance for chronic VEEV disease. Aside from some minor grammatical errors and omission of detail on the control animals the manuscript is well written and discussed; however, it is this reviewers opinion that the manuscript would benefit from an assessment of the impact of gender/age on the results even if minor or inconclusive.

Reviewer #3: This manuscript offers new insights into VEEV infection in nonhuman primates exposed to small particle aerosols. The strengths of this study lie in using an infectious clone of VEEV and including telemetry data to track neurological abnormalities that are sometimes absent by cage-side observation alone or traditional data acquisition such as blood work and virologic analyses. However, unfortunately the manuscript lacks detail and could be executed in a better way with regard to attention to detail, writing, material references and discussing the findings within a broader and more relevant context.

**Part II – Major Issues: Key Experiments Required for Acceptance**

Reviewer #1: 1. Key details regarding methodology are lacking. Please review the Materials & Methods section and ensure that sufficient details are provided. Some specific examples that should be addressed: the section on cells/media are missing; details regarding the aerosol exposure such as maximal exposure time, sampling efficiency of the AGI for VEEV, location of sampling for determination of the exposure dose (methods say chamber, but is it at the point of the head only device?). For EEG/ICP/Temp: how long was data acquisition stopped and was this the same time every day?

2. For clinical scoring the authors indicate that they increased observations when NHPs reached severe disease (score of 10) and monitored animals every 8 hrs, including overnight. Given that they were also monitoring circadian patterns, how did they account for the overnight disturbances in the circadian pattern data when observations were increased as this would likely create noise in the data.

3. The method section for pathology requires more detail. How long were samples fixed in formalin prior to removal from biocontainment? Provide a reference or detail for tissue/slide preparation. Were samples scored by a boarded pathologist?

4. The pathology section needs more detail to be convincing. The main conclusion based on path results is that there is evidence of diffuse inflammation throughout CNS regions after recovery from physiological responses. Based on the data, it appears that the majority of animals had mild to no lesions (Fig 13G). The H&E images appear to be focal lesions, but due to the increased magnification of the panels from left to right, appear more diffuse. The other important question that was not addressed was whether the areas of inflammation observed by histopath correspond to either vRNA or local cytokine production. Immunostaining would be far more convincing here.

5. In looking at Fig 1, it is difficult to appreciate a biphasic fever. The low dose group is perhaps the most convincing (with the exception of NHP 6-19). In general, the fever response appears more undulating with multiple peaks and valleys. Can the analysis for delineating the fever phases be described in more detail with respect to the statistical analysis that was used to determine if data points represented a unique phase. What is the definition of fever in the model (x degrees above baseline)?

6. Could the 'noise' in Fever 2 be due to it being more than a single phase and actually multiple fevers during that time?

7. Figure 5: Since the significant difference in circadian index is seen in F2, could this be due to the fact that you increased observations during that time period and disturbed the natural sleep cycle of the animals (ie: is this artificially induced)?

8. what is the background in Fig 8 A & C that is subtracted? It doesn't make sense to have such low titers by PCR relative to the infectious titers shown.

9. The sequence of the infectious clone virus stock should be provided particularly since the authors correctly point out that prior studies used cell culture passaged virus that was not genotyped. This paper should indicate in more detail how the virus was rescued and number of passages that were required to obtain the stock that was characterized and used for aerosol challenge.

10. The authors mention the significance of a biphasic fever in human cases of VEEV, however, in recent reviews it was shown that the observation of biphasic fever was actually not a common occurrence. Can the authors comment on this? Reference = PMID 30282570

11. A tremendous amount of work and data are shown with respect to inflammatory mediators. The discussion could be expanded further to put these data in context with the other data that are presented.

Reviewer #2: None.

Reviewer #3: Though no additional experiments are suggested, careful and extensive re-writing of the text and re-framing of the findings are required. Please see extent of comments in next section.

**Part III – Minor Issues: Editorial and Data Presentation Modifications**

Reviewer #1: 1. Figure 13: Text refers to 13G, however, the chart in Fig 13 is not labelled

2. There are several broad/general comments (particularly in the introduction) that require clarification and references. Specifically, the first sentences in paragraphs #1 and #2 as they imply that aerosol transmission is a typical route of transmission (para 1) and is highly infectious in humans via aerosol (para 2). I suspect the authors are referring to animal studies that have shown this to be the case.

3. Under challenge virus in M&M, the last sentence refers to "dose-step infection of CD-1 mice". The reviewer assumes this was an LD50 study. What is the "typical virulence" the authors refer to? Is this the known LD50 in mice? Please be more specific.

Reviewer #2: The purpose of the study in the last paragraph of the introduction could be written better for clarity and EEG and ICP should be written out at first introduction.

Details on the mock/uninfected animals should be included in the methods.

SIV, STLV and SRV should be spelled out under general animal procedures.

Aerosol Exposure "attached to the chamber and operated at...

EEG data analysis - be consistent in how the HZ is presented and the use of parenthesis

Brain cell isolation = centrifuged at 500 x gravity

RNA isolation = phase was collected then combined

In the results section the Johnston Ref should be detailed in the first paragraph and it's worthwhile to state that regardless of the dose that exposure to aerosolized VEEV was non-lethal during the period examined.

Need to be consistent with use of average vs median delta T in the second paragraph of the fever response.

Provide a description of mock infected animals in results and be consistent when referring to mock infected vs. uninfected. These were the same? Was the temperature of mock infected animals observed? Was telemetry done on those animals as well or does this data come from a previous study.

Were there any gender or age differences in the results? This is important for NIAID/FDA.

Second paragraph of discussion TrD, which was originally isolated ...

Discuss gender/age differences even if nothing significant was observed

Most of the figures compare mock vs VEEV in that order with the exception of Figure 11. Suggesting making this consistent throughout.

Reviewer #3: General:

o Remove double spaces throughout manuscript

o Replace “infection” with “exposure”

Abstract:

o Spell out infectious clone (IC)

o Human infection should be replaced with human case of VEEV

o Replace CNS with CSF or CNS tissues

o Replace “viral” with “VEEV” genomic nucleic acid

o Rephrase “persistence of virus replication” with “persisting virus” unless authors have evidence of active replication such as isolation of infectious VEEV

Introduction

o Consider splitting up the first sentence or leave out “cause disease when inhaled” and explain in subsequent paragraph that the inhalational route is of relevance with regard to biothreat purposes.

o Replace “instrumental” with “important study” or another more suitable statement

o Explain why crab-eating macaques are useful, or relevant models of alphavirus infection. Why was not a mouse model used for the study, especially as wildtype VEEV is more lethal pin mice than in nonhuman primates as the authors state?

o Spell out “FDA”

o What do the authors consider as “low passage” virus stock? Are genetic changes acquired during cell culture passage of virus, and if so do they contribute to phenotypic changes as well (for example attenuation to in vivo)?

o Spell out “NHP”

o In the last paragraph the authors state that disease in NHPs has only been characterized observationally, but then first objective of this study was to further characterize disease which is a contradiction

Material and Methods

o Please add the ALAAC and PHS accreditation and approval numbers specific to your institution

o Challenge virus: Was one or multiple strains used? If exposure was only performed with one infectious clone, then please state accordingly

o Have the authors confirmed the sequence of the infectious clone after cell culture passage and prior to confirmation of virulence in mice to ensure that no additional mutations were acquired through propagation in Vero cells?

o Correct 37C to 37 °C

o What was the cut-off for the PRNT, was it 50% or 80%? Also, please add material references for the commercial mouse antibody

o Add a space between “200” and “mg/kg”

o What is “Beuthanasia”? Is it pentobarbital? Please reference any commercial product used and add the dosage.

o Was euthanasia or death confirmed? How was death confirmed?

o Change “tracheal tube” to “endotracheal tube”

o How long were the aerosol exposures? Was this a time-calculated run or accumulated dose based on respiration?

o What is lpm? Do the authors mean 16 liters per minute? Please clarify

o What was the duration of particle size measurements? It was started at 5 minutes into the exposure and lasted until which minute? Was it the same across all sprays?

o How was baseline behavior of nonhuman primates normalized to behavior recorded post-exposure?

o The clinical scoring example in parenthesis is not necessary, please remove. Is it clear that a cumulative score of 10 triggered euthanasia, but please indicate if secondary parameters were used to meet the euthanasia criterion (such as blood parameters like glucose or leucocytes, or extreme hypothermia).

o How was plasma obtained? Please add blood tubes, centrifugation details etc.

o Please list all tissues collected during necropsy including the ones from different regions of the brain, was the collection different between nonhuman primates? Were different tissues collected from controls?

o The nucleic acid isolation and PCR sections are missing material references.

o Please remove “in-house”

o How many 10-fold dilutions were used to generate a standard curve for quantification of gene copies by RT-PCR?

o

Results

o Are the delivered dose ranges that the authors used for grouping of the exposed nonhuman primates biologically relevant?

o How did the authors determine fever? What were the core temperatures measured during the febrile periods?

o How were significant differences in clinical signs between the exposed groups assessed?

o What does it mean if there is “aggregate decrease in all wavebands”?

o Please rephrase: “As has been reported previously with VEEV infection of non-human primates (5, 6), after aerosol exposure to INH-9813, there was a significant drop in white blood cell counts of macaques in the first five days after infection, and most of that loss was in the lymphocyte population.

o Please use more accurate terms within the hematology section and include results for glucose.

o Viremia section: This section lacks detail, especially to the level of virus detected, please add how much virus was detected (titer and VEEV copies/mL)

o Were increases in cytokines significantly higher in nonhuman primates with higher viral load and if so, what is the level of significance?

o Please explain what presence of the three markers of traumatic brain injury suggests

Discussion

o How specifically are the telemetry data useful and informative for future evaluation of countermeasures. What if use of telemetry and implantation are not possible? Is there another surrogate biomarker that could bridge this gap?

o Please explain what low- and high-passage virus stock means in the context of virulence studies. What are changes of concern? Was the infectious clone sequenced in the current study?

o Please list and discuss limitations of your study.

o Please discuss implications and benefit of your findings towards the improvement of future studies.

PLOS authors have the option to publish the peer review history of their article (what does this mean?). If published, this will include your full peer review and any attached files.

Reviewer #1: No

Reviewer #2: No

Reviewer #3: No
---

## [Decision Letter · Decision Letter 1]

7 Feb 2022

Dear Dr. Reed,

Thank you very much for re-submitting your manuscript "Long-term persistence of viral RNA and inflammation in the CNS of macaques exposed to aerosolized Venezuelan equine encephalitis virus." for consideration at PLOS Pathogens. As with all papers reviewed by the journal, your manuscript was reviewed by members of the editorial board and by several independent reviewers. In light of the reviews (below this email), we would like to invite the resubmission of a significantly-revised version that takes into account the reviewers' comments.

Specifically, in-depth stock/exposure virus characterization should be performed/included in the manuscript.

We cannot make any decision about publication until we have seen the revised manuscript and your response to the reviewers' comments. Your revised manuscript is also likely to be sent to reviewers for further evaluation.

Sincerely,

Jens H. Kuhn

Associate Editor

PLOS Pathogens

Mark Heise

Section Editor

PLOS Pathogens

Kasturi Haldar

Editor-in-Chief

PLOS Pathogens

orcid.org/0000-0001-5065-158X

Michael Malim

Editor-in-Chief

PLOS Pathogens

orcid.org/0000-0002-7699-2064

Reviewer's Responses to Questions

**Part I - Summary**

Reviewer #1: The authors have addressed the majority of all reviewer comments, however, several key shortfalls still remain. In a number of cases, rather than directly addressing questions it was noted that language was just softened. This missed the mark and did not fully address the issues. A specific example was fever response.

Reviewer #2: In this revised manuscript, the authors have adequately addressed reviewer comments to improve the interpretation of the data and support the conclusions being drawn.

Reviewer #3: (No Response)

**Part II – Major Issues: Key Experiments Required for Acceptance**

Reviewer #1: 1. The authors responded to comment 1 by stating, "Sampling efficiency was not empirically determined in this study as it is

not relevant to the inhaled dose; in prior studies the sampling efficiency for the AGI we use is approximately 97%." The reviewer disagrees that this is not relevant as it will ultimately influence the amount of virus that can be recovered and estimate of the dose. Given that the aim was to use a well characterized stock and further characterize the model, this is a major point that should be empirically tested. Recovery efficiencies will vary by agent and can contribute to differences observed in animal models if it is not well characterized prior to challenge.

2. The fact that the viral stock that was used in the challenge was not well characterized remains a major concern. This material should be fully sequenced and characterized. Even a single passage in cell culture can result in SNPs that may or may not influence pathogenesis. While confirmation of virulence in the mouse model does alleviate some concern, it doesn't necessarily suffice for proper stock characterization. Given that the authors point out the importance of this, it is difficult to justify publication without this.

Reviewer #2: None

Reviewer #3: (No Response)

**Part III – Minor Issues: Editorial and Data Presentation Modifications**

Reviewer #1: (No Response)

Reviewer #2: In the description of viremia viral RNA titers should be reported as copies/ml as opposed to pfu/ml.

The added discussion on gender differences (page 46) does not make sense and should be rewritten for clarity.

the abbreviation for EEEV has been introduced previously so it is not necessary to write out eastern equine encephalitis virus on page 49

Reviewer #3: Given the potential impact of single nucleotide mutations on the phenotype of the viral stock used, and the author's discussion of low versus high passage stocks used for aerosol exposure, it is important to provide confirmation that the Vero cell passaged stock used for the NHP exposure matches the reference sequence. A phenotypic confirmation in mice or in vitro is not sufficient. Please provide confirmation of the sequence in your response.

PLOS authors have the option to publish the peer review history of their article (what does this mean?). If published, this will include your full peer review and any attached files.

Reviewer #1: No

Reviewer #2: No

Reviewer #3: No
---

## [Editor Report · Decision Letter 2]

11 May 2022

Dear Dr. Reed,

We are pleased to inform you that your manuscript 'Long-term persistence of viral RNA and inflammation in the CNS of macaques exposed to aerosolized Venezuelan equine encephalitis virus.' has been provisionally accepted for publication in PLOS Pathogens.

Best regards,

Jens H. Kuhn

Associate Editor

PLOS Pathogens

Mark Heise

Section Editor

PLOS Pathogens

Kasturi Haldar

Editor-in-Chief

PLOS Pathogens

orcid.org/0000-0001-5065-158X

Michael Malim

Editor-in-Chief

PLOS Pathogens

orcid.org/0000-0002-7699-2064
---

## [Editor Report · Acceptance letter]

7 Jun 2022

Dear Dr. Reed,

We are delighted to inform you that your manuscript, "Long-term persistence of viral RNA and inflammation in the CNS of macaques exposed to aerosolized Venezuelan equine encephalitis virus.," has been formally accepted for publication in PLOS Pathogens.

Best regards,

Kasturi Haldar

Editor-in-Chief

PLOS Pathogens

orcid.org/0000-0001-5065-158X

Michael Malim

Editor-in-Chief

PLOS Pathogens

orcid.org/0000-0002-7699-2064